# An IL-10/DEL-1 axis supports granulopoiesis and survival from sepsis in early life

Eleni Vergadi ®[1,2,9] ✉, Ourania Kolliniati[2,3,9], Ioanna Lapi[2,3],
Eleftheria Ieronymaki[2,3], Konstantina Lyroni ®[2,3], Vasileia Ismini Alexaki[4],
Eleni Diamantaki ®[5], Katerina Vaporidi[5], Eleftheria Hatzidaki ®[6],
Helen A. Papadaki[7], Emmanouil Galanakis[1], George Hajishengallis ®[8],
Triantafyllos Chavakis ®[4] & Christos Tsatsanis ®[2,3]

The limited reserves of neutrophils are implicated in the susceptibility to infection in neonates, however the regulation of neutrophil kinetics in infections in early life remains poorly understood. Here we show that the developmental endothelial locus (DEL-1) is elevated in neonates and is critical for survival from neonatal polymicrobial sepsis, by supporting emergency granulopoiesis. Septic DEL-1 deficient neonate mice display low numbers of myeloid-biased multipotent and granulocyte-macrophage progenitors in the bone marrow, resulting in neutropenia, exaggerated bacteremia, and increased mortality; defects that are rescued by DEL-1 administration. A high IL-10/IL-17A ratio, observed in newborn sepsis, sustains tissue DEL-1 expression, as IL-10 upregulates while IL-17 downregulates DEL-1. Consistently, serum DEL-1 and blood neutrophils are elevated in septic adult and neonate patients with high serum IL-10/IL-17A ratio, and mortality is lower in septic patients with high serum DEL-1. Therefore, IL-10/DEL-1 axis supports emergency granulopoiesis, prevents neutropenia and promotes sepsis survival in early life.

Sepsis remains a serious threat in early life, leading to significant morbidity and mortality[1–3]. Newborn's susceptibility to infection is mainly attributed to immature immune responses and/or limited reserves of neutrophils in the bone marrow[4–7], however, the underlying mechanisms that regulate neonatal immunity remain poorly understood.

Neutrophils are an essential first line of innate immune response against invading pathogens[8,9]. Under resting conditions, neutrophils are retained in the bone marrow and circulation, while, during infection, inflammatory mediators stimulate their release from the bone marrow and their translocation from the circulation to the inflamed

tissues[9]. In early life, the bone marrow neutrophil storage pool and its ability to increase neutrophil production are limited[10–12]. Furthermore, neutrophil recruitment to inflamed tissues is reduced in newborns compared to adults[10–12]. In this regard, newborns display both quantitative and qualitative neutrophil defects and, consequently, often exhibit neutrophil bone marrow depletion, peripheral neutropenia and inadequate neutrophil tissue infiltration upon infection[11]. Nevertheless, tight regulation of neutrophil kinetics is vital in the neonatal host defense.

The developmental endothelial locus-1 (DEL-1), encoded by the *EDIL3* gene, is a soluble endogenous protein secreted by endothelial

[1]Department of Paediatrics, School of Medicine, University of Crete, Heraklion, Greece. [2]Institute of Molecular Biology and Biotechnology, IMMB, FORTH, Heraklion, Greece. [3]Department of Clinical Chemistry, School of Medicine, University of Crete, Heraklion, Greece. [4]Institute for Clinical Chemistry and Laboratory Medicine, University Hospital and Faculty of Medicine, Technische Universität Dresden, Dresden, Germany. [5]Department of Intensive Care Medicine, School of Medicine, University of Crete, Heraklion, Greece. [6]Department of Neonatology/Neonatal Intensive Care Unit, School of Medicine, University of Crete, Heraklion, Greece. [7]Department of Hematology, School of Medicine, University of Crete, Heraklion, Greece. [8]Department of Basic and Translational Sciences, School of Dental Medicine, University of Pennsylvania, Philadelphia, PA, USA. [9]These authors contributed equally: Eleni Vergadi, Ourania Kolliniati. ✉e-mail: eleni.vergadi@uoc.gr

and other tissue-resident cells, which binds to leukocyte β2 integrins (such as the lymphocyte function-associated antigen, LFA-1 or cluster of differentiation, CD11a/CD18) and antagonizes their interaction with the intracellular cell adhesion molecule 1 (ICAM-1) on the endothelium[13–17]. By regulating the LFA-1–ICAM-1 interaction, the endothelial cell-derived DEL-1 can control leukocyte recruitment to peripheral tissues and thus the extent of local inflammation[14]. Moreover, macrophage-derived DEL-1 promotes phagocytosis of apoptotic neutrophils (efferocytosis) and reprogramming of macrophages to a pro-resolving phenotype, thereby contributing to successful resolution of inflammation[18]. DEL-1 counteracts interleukin (IL)−17A-dependent inflammation[19,20]. Consequently, DEL-1 deficiency is associated with increased susceptibility to a variety of inflammatory conditions, especially those driven by IL-17A[21,22]. DEL-1 has been also shown to interact with β3 integrin (CD61) on hematopoietic stem cells (HSCs) and induce HSC proliferation and differentiation toward the myeloid lineage in the bone marrow[23]. In addition, DEL-1 regulates emergency myelopoiesis in response to granulocyte colony-stimulating factor (G-CSF) or upon lipopolysaccharide (LPS) - driven inflammation[23]. DEL-1 expression is downregulated in peripheral tissues or organs upon inflammation, for instance by the pro-inflammatory cytokines tumor necrosis factor (TNF) and IL-17A in adult animal models of inflammatory diseases[20,24]. While DEL-1 is known to be suppressed upon pro-inflammatory stimuli, essentially no information exists on the role of anti-inflammatory cytokines, such as IL-10, on DEL-1 expression.

Despite the wealth of information on the role of DEL-1 in adult mouse models of inflammation, no information is available on the expression and functional significance of DEL-1 in sepsis in early life. The expression of DEL-1 in neonates and its effect on the outcome of neonatal innate immune responses and neutrophil kinetics upon sepsis have not been hitherto evaluated. Given the substantial implications of newborn infections, a better understanding of the mechanisms regulating inflammation and neutrophil circulation in the neonatal period is pivotal to improve survival in these susceptible young hosts. Herein, we aimed to shed light on the expression and function of DEL-1 in neonate hosts and evaluate its potential prognostic and/or therapeutic value in the management of neonatal sepsis.

In this study, we show that healthy neonate pups exhibited higher levels of tissue DEL-1 compared to adults. Upon cecal slurry-induced polymicrobial sepsis (CS sepsis), DEL-1 levels were downregulated in adult mice, but remained unaffected in neonates, who also exhibited reduced neutrophil tissue infiltration compared to adults. *Edil3⁻/⁻* (hereafter designated *Del1⁻/⁻*) neonate mice demonstrated enhanced neutrophil tissue infiltration but soon developed neutropenia in the blood as well as the bone marrow and exhibited worse survival in sepsis, when compared to DEL-1-sufficient neonate mice. Interestingly, DEL-1 levels were elevated in the bone marrow of wild type (WT) neonate mice compared to adults. DEL-1 was essential to promote emergency granulopoiesis and facilitate sustained output of circulating neutrophils, thereby controlling bacteremia and survival from sepsis. We also demonstrate that the expression of DEL-1 and its effects on neutrophil production are under the control of IL-10. Together, these findings highlight a significant role of an IL-10/DEL-1 axis in preventing sepsis-induced neutropenia and promoting survival from sepsis in neonates.

## Results

### Expression of the homeostatic factor DEL-1 is elevated in early life

To determine whether tissue expression of DEL-1 differs between adults and neonates, we analyzed different tissues from healthy adult and neonate mice. DEL-1 mRNA was highly expressed in murine neonatal brain and lung tissue, to a lesser degree in intestine and kidney, while very low expression of the DEL-1 transcript was observed in the heart, liver and spleen (Fig. 1a). We compared the DEL-1 mRNA

expression in the lung, kidney, intestine, and brain of neonate (4 days old) and adult (8–10 weeks old) mice (Fig. 1b). DEL-1 mRNA expression was higher in neonate mouse pups (4 days old) compared to adult mice in all tissues studied (lung, kidney, intestine) except for the brain (Fig. 1b). As postnatal age advanced, DEL-1 mRNA expression was gradually reduced in tissues that were examined (lung, intestine and kidney) (tested at day 1, 7, 14 and 21 of life in mouse pups as well as in adult mice 8–10 weeks old) (Fig. 1c–e). We also measured human DEL-1 protein in cord blood serum from newborns [gestational age (GA) 34 to 40 weeks], peripheral blood serum from children at the age of 4 years (Fig. 2a) as well as healthy newborns and healthy adults (Fig. 2b). Median DEL-1 serum concentration was significantly higher in neonates compared to older children (Fig. 2a) and adults (Fig. 2b).

The expression of ICAM-1 and β2 integrin LFA-1 (CD11a/CD18), one of the DEL-1 receptors, were also evaluated in neonatal tissues and neutrophils, respectively. The mean fluorescence intensity of CD11a protein in blood neutrophils did not differ between adult and neonate mice (Supplementary Fig. 1a, b). Additionally, there was no difference in ICAM-1 mRNA expression in various tissues from healthy adult and neonate mice (4 days old) (Supplementary Fig. 1c).

### DEL-1 expression is not suppressed upon sepsis in neonates

Earlier studies have shown that DEL-1 is suppressed upon acute inflammation in several animal models of disease and may resurge in the context of resolution of inflammation[14,18,20,25,26]. The regulation of DEL-1 expression during neonatal sepsis remains unknown. To address this, we subjected WT adult and neonate mice to the cecal slurry (CS) model of polymicrobial peritonitis, a gold standard model for neonatal sepsis studies[27].

Following intraperitoneal CS administration, mRNA expression of DEL-1 was determined at different time points in the lung and intestine tissues of neonate pups and adult mice, representing the site of infection (intestine) and a peripheral organ, such as lung, where we showed that DEL-1 is substantially expressed (Fig. 1a). DEL-1 mRNA was downregulated 6 hours after CS-induced sepsis in adults and returned to baseline at 12 hours, while it was elevated 6 hours after CS-induced sepsis in neonate pups (Fig. 3a). DEL-1 protein in lung tissue lysates was also elevated in septic neonate pups compared to adult mice (Fig. 3b). We also compared changes of DEL-1 expression in intestine 6 hours after CS-induced sepsis. DEL-1 was suppressed in the intestine of adult mice, while in neonates it remained unchanged (Fig. 3c). DEL-1 was detectable in human serum where a similar pattern was observed; DEL-1 was suppressed in septic adults (during the first 24 hours of sepsis) compared to healthy controls, while such a decrease was not evident in neonates with sepsis during the same time interval (Fig. 2b). Protein expression of the DEL-1 receptor, CD11a, was upregulated similarly in the blood neutrophils of both neonate and adult mice with sepsis (Supplementary Fig. 1a, b). ICAM-1 mRNA expression was also similarly upregulated in both neonate and adult septic mice (Supplementary Fig. 1d).

### DEL-1 controls tissue neutrophil infiltration in neonates

DEL-1 regulates neutrophil recruitment to the site of inflammation. We therefore evaluated potential differences in neutrophil infiltration in the peritoneal cavity and lung of neonate and adult mice exposed to CS-induced sepsis, via measuring absolute neutrophil numbers in the peritoneal cavity and myeloperoxidase (MPO) activity in lung tissue lysates. MPO has been shown to be equally expressed in neutrophils of term neonates and adults[28], thus it is a reliable marker of neutrophilic infiltration in both adult and neonate mice.

We found that septic neonate pups exhibited lower numbers of neutrophils (CD11b⁺Ly6G⁺) in peritoneal lavage fluid and reduced MPO activity in the lung compared to adults (Fig. 3d, e). No difference in CD11b⁺Ly6C⁺Ly6G⁻ monocyte population in the peritoneal lavage was noted between healthy and septic WT and *Del1⁻/⁻* mice (Supplementary

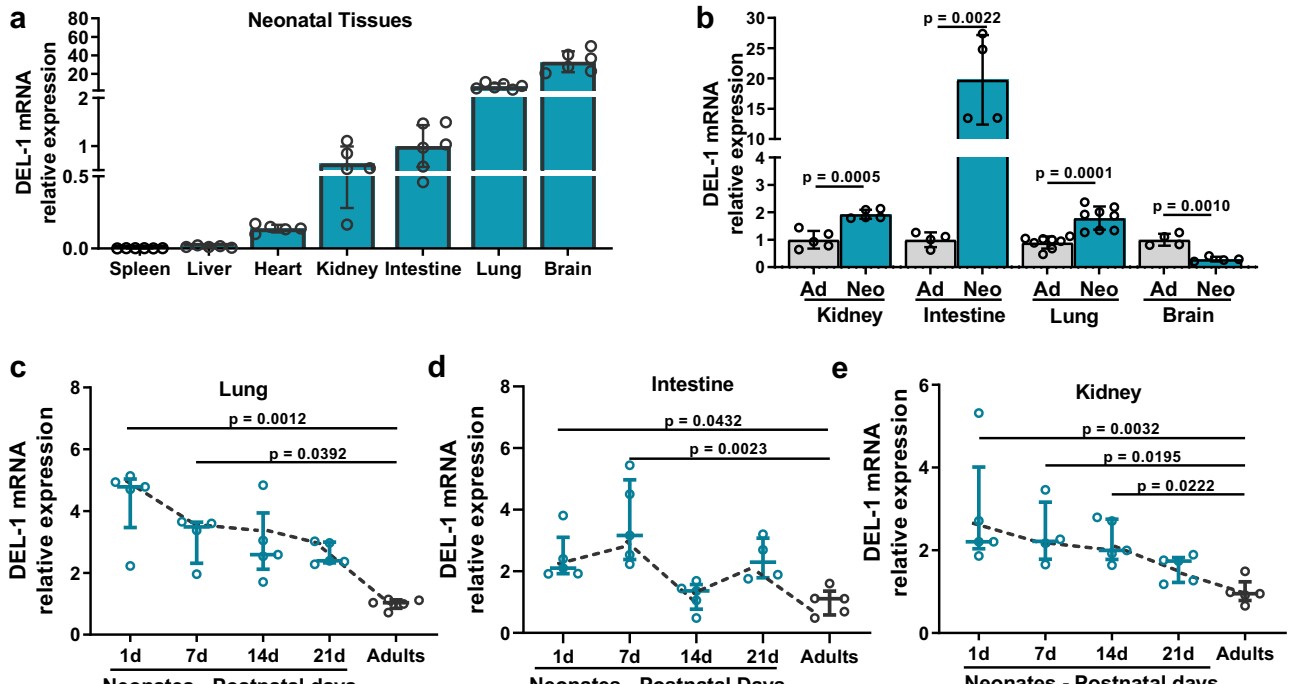

**Fig. 1 | Tissue DEL-1 expression is elevated in neonatal age. a** DEL-1 relative mRNA expression in different tissues of healthy C57BL/6 neonate mice (4 days old) (expression of DEL-1 in the intestine was set as 1) (*n* = 5 animals per group in heart, liver and kidney and *n* = 6 animals per group for the rest of the groups). **b** DEL-1 relative mRNA expression in tissues from healthy C57BL/6 mice of adult (8–10weeks) or neonatal age (4 days old) (expression of DEL-1 in each tissue of adult mice was set as 1) (*n* = 5 animals per group in adult and neonatal kidney groups, *n* = 4 animals per group in adult and neonatal intestine groups, n = 8 animals per in adult and neonatal lung group and *n* = 4 animals per group on adult and neonatal brain tissue groups). DEL-1 relative mRNA expression in the **c** lung (*n* = 5 animals per group in adult and 1d, 14d and 21d neonate groups, n = 4 animal in 7d neonate group), **d** intestine (*n* = 5 animals per group in adult and 1d, 7d, and 14d neonate groups, *n* = 4 animal in 21d neonate group) and **e** kidney tissue (*n* = 5 animals per group in adult and 1d, 14d, and 21d neonate groups, *n* = 4 animal in 7d neonate group), from healthy C57BL/6 adult mice and healthy neonate mice of 1, 7, 14 or 21 days of postnatal age (the expression of DEL-1 in each adult tissue was set as 1). Statistical analysis by two-sided unpaired *t* test (**b**) and Kruskal Wallis with Dunn's multiple comparison post-test (**c**–**e**). Mean ± SD (**a**, **b**) and median ± interquartile range (**c**–**e**) are depicted. Source data are provided as a Source Data file. Ad adults, Neo neonates, d days.

Fig. 2a). Thus, higher levels of DEL-1 in neonates, as compared to adults, are associated with reduced tissue neutrophil recruitment.

To evaluate the impact of DEL-1 on neutrophil recruitment in neonates, we subjected WT and *Del1*−/− neonate mouse pups to CS-induced sepsis and evaluated neutrophil recruitment to the peritoneum and lung. *Del1*−/− septic neonate mice had elevated numbers of neutrophils in peritoneal lavage fluid (Fig. 3f) and higher MPO activity in the lung (Fig. 3f), compared to WT septic mouse pups. In contrast, septic *Del1*−/− neonate mice that received i.v. recombinant DEL-1 protein fused with human IgG-Fc (DEL-1-Fc), had reduced numbers of neutrophils in peritoneal lavage fluid and reduced MPO activity in the lung compared to septic *Del1*−/− neonates that received control IgG-Fc (Fig. 3f). However, *Del1*−/− neonate mice treated with DEL-1-Fc still had significant neutrophilic infiltration, similar to that of WT mice, suggesting that DEL-1-F did not completely abolish but mainly suppressed the exaggerated neutrophil infiltration of septic *Del1*−/− neonate mice.

## DEL-1 promotes sepsis survival in neonates by preventing peripheral blood neutropenia

Next, we evaluated the effect of DEL-1 deficiency on survival from CS-induced sepsis in neonate pups. *Del1*−/− neonate pups exhibited reduced survival upon sepsis compared to WT counterparts when sepsis was mild or moderate, while at severe sepsis the difference was no longer evident (Fig. 4a–c), suggesting that endogenous DEL-1 confers protection against sepsis.

To delineate the mechanism of DEL-1-mediated protection against neonatal sepsis, we evaluated cytokine responses, neutrophil

infiltration, phagocytosis capacity, and bacteria clearance in WT and *Del1*−/− neonate mice in the presence or absence of exogenous DEL-1 administration using DEL-1-Fc[21,24]. No difference was observed in serum TNF and IL-10 protein concentration between WT and *Del1*−/− neonate pups (Supplementary Fig. 3a, b) or between *Del1*−/− neonate pups treated either with i.v. DEL-1-Fc or control IgG, 6 hours following CS-induced sepsis (Supplementary Fig. 3c, d).

Administration of a broad-spectrum antibiotic significantly reduced the mortality rate of septic *Del1*−/− neonate mice (threefold), while to a lesser degree in WT mice (Supplementary Fig. 4). This finding suggests that *Del1*−/− mice have exaggerated bacterial load, due to either reduced phagocyte numbers or defective phagocytosis capacity, that contributes to increased mortality. Indeed, although at 6 hours following initiation of septic *Del1*−/− mice had less or similar bacterial burden in the peritoneum and blood, respectively, at 12 hours post-sepsis, they exhibited higher bacterial load in the blood, as compared to WT neonate mice (Fig. 4d, e).

To investigate whether the presence or absence of DEL-1 could have affected phagocytosis capacity in our model, we performed an ex vivo blood neutrophil phagocytosis assay in neutrophils derived from WT mice of adult or neonatal age. Term neonates are partially deficient in opsonins, such as complement and IgG, but in absence of opsonins they are known to have equal phagocytosis activity compared to adults[29–31]. To recapitulate the phagocytosis conditions in early life, we infected neutrophils ex vivo, in the presence or absence of DEL-1-Fc, with unopsonised FITC-labeled *Escherichia coli (E. coli)*. DEL-1-Fc administration, as compared to IgG-Fc control, did not affect the ex vivo blood phagocytosis capacity of non-opsonized FITC-

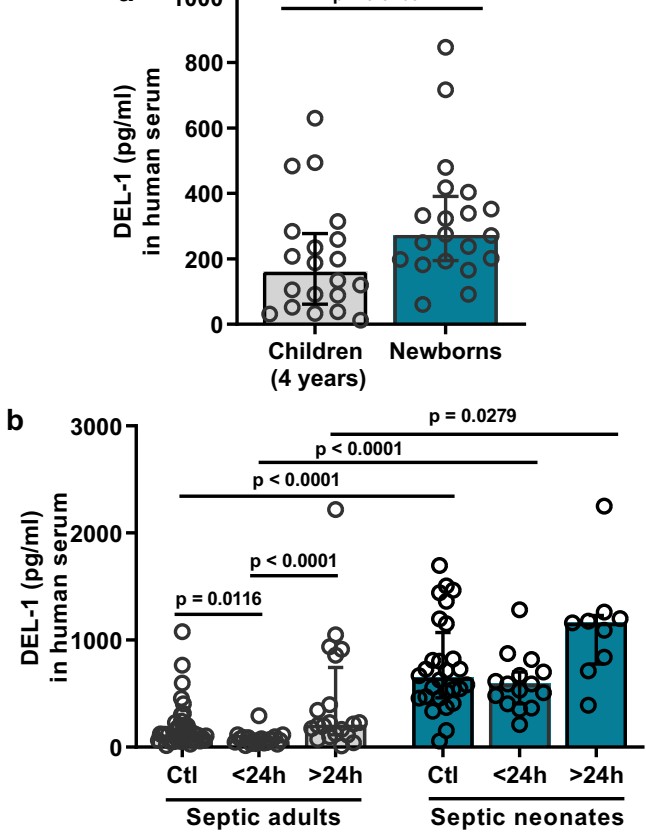

**Fig. 2 | DEL-1 expression in human neonates and adults. a** Median human DEL-1 protein concentration in cord blood serum from healthy newborns (gestational age 34–40 weeks old) and from healthy children at the age of 4 years old (*n* = 20 children and *n* = 20 newborns). **b** Median human DEL-1 protein serum concentration from healthy and septic adults and neonates (within <24 hours and >24 hours of sepsis onset) (*n* = 40 adult controls, *n* = 20 adults in <24 hours sepsis group, *n* = 20 adults in >24 hours sepsis group, *n* = 28 neonate controls, *n* = 15 neonates in <24 hours sepsis group and *n* = 9 neonates in >24 hours sepsis group). Statistical analysis by two-sided Mann-Whitney test (**a**) and Kruskal-Wallis test with Dunn's multiple comparison post-test. Median ± interquartile range is depicted (**a**, **b**). Source data are provided as a Source Data file. Ctl control, h hours.

*E. coli* particles by either adult or neonatal neutrophils (Supplementary Fig. 5).

The time course of neutrophil abundance in the peritoneum and blood followed the reverse pattern of that of bacterial load. Specifically, at 6 hours following initiation of sepsis, *Del1*−/− mice had initially higher or similar neutrophil numbers in the peritoneum and blood, respectively, but the number of neutrophils decreased significantly at 12 hours post-sepsis and were significantly lower as compared to WT mice (Fig. 4f, g), suggesting that *Del1*−/− mice suffer from significant neutropenia during the course of severe sepsis.

Moreover, DEL-1-Fc administration in the context of severe sepsis significantly improved the survival of *Del1*−/− septic neonate pups (Fig. 4h). The increased bacterial load associated with DEL-1 deficiency was reversed by DEL-1-Fc administration both in blood circulation (Fig. 4i) and at the site of infection (peritoneal cavity) (Supplementary Fig. 6). DEL-1-Fc administration prevented the decline in neutrophil numbers in DEL-1 deficient neonate pups (Fig. 4j).

To address the relevance of the above murine findings in humans, we evaluated the 28-day mortality rate in adult and neonate patients with sepsis that had either low or high serum DEL-1 concentration. Based on the median DEL-1 concentration of healthy sex- and age-matched humans that we included in this study (Fig. 2), the threshold

for high/low DEL-1 concentration was determined to 125 pg/ml for adult patients and 700 pg/ml for neonates. The clinical characteristics, source of sepsis, and outcome of the two groups of neonates and adults are depicted in Supplementary Table 1 and 2, respectively. There were no differences in demographic characteristics or type/source of sepsis between low or high serum DEL-1 neonates or adults with sepsis (Supplementary Table 1 and 2). Neonates with sepsis in the group with high serum DEL-1 levels appeared to exhibit lower 28-day mortality rate compared to the group with low serum DEL-1 levels (8.3% vs. 25%, respectively) (Fig. 5b and Supplementary Table 1). Furthermore, adult patients with sepsis in the group with high serum DEL-1 exhibited lower 28-day mortality rate compared to the group with low serum DEL-1 levels (13.3% vs 44%, respectively, *p* < 0.05) (Fig. 5c and Supplementary Table 2). Consistent with the above findings, neonate patients with sepsis that had high serum DEL-1 concentration on the first day of enrollment, exhibited a significant increase in the blood neutrophil count (twofold) the following 24 hours compared to the neonates of the group with low serum DEL-1 concentration (Fig. 5a and Supplementary Table 1).

### DEL-1 is elevated in the neonatal bone marrow and supports the bone marrow neutrophil pool upon sepsis
The observation that *Del1*−/− neonate mice failed to sustain circulating neutrophils in the blood early upon sepsis, prompted us to further investigate whether this phenomenon is attributed to deficits in neutrophil bone marrow production and release in *Del1*−/− neonate mice. To address this hypothesis, we first evaluated DEL-1 expression and the number of mature neutrophils (CD11b+Ly6G+Ly6C−) in the bone marrow of healthy and septic WT and *Del1*−/− mice of neonatal age and compared them to adult WT mice.

While DEL-1 mRNA and protein expression in the bone marrow were not significantly altered upon sepsis in adult mice, they were significantly increased 12 hours after sepsis in neonate mice (Fig. 6a, b), suggesting that DEL-1 may have a significant regulatory role in the neonatal bone marrow niche. In addition, WT neonate mice had a smaller pool of neutrophils in the bone marrow both under steady-state and septic conditions compared to WT adult mice (Fig. 6c, d). Compared to WT neonates, *Del1*−/− neonates had similar numbers of neutrophils (CD11b+Ly6C−Ly6G+) in the bone marrow at steady-state but significantly lower during 12 hours of sepsis (Fig. 6c). We also observed that the bone marrow pool of neutrophils in *Del1*−/− neonates decreased over time as sepsis progressed (Fig. 6c). Moreover, DEL-1-Fc administration increased the number of neutrophils in the bone marrow of *Del1*−/− septic neonate pups (Fig. 6d).

No difference was noted in other cell populations in the bone marrow, such as the CD11b+Ly6C+Ly6G− monocyte population or myeloid-derived suppressor cells (MDSCs) (CD11c−CD11b+ Ly6G+ Ly6C+) between WT and *Del1*−/− neonate mice (Supplementary Figs. 2b and 7).

### DEL-1 supports emergency granulopoiesis in the bone marrow of neonates
Since we observed a significantly reduced neutrophil pool in the bone marrow of *Del1*−/− mice, we aimed to further characterize the role of DEL-1 in bone marrow granulocyte production and release.

First, we assessed whether the decreased numbers of granulocytes in the bone marrow of *Del1*−/− mice was associated with altered expression of stromal cell-derived factor 1 (SDF-1) in the bone marrow, a soluble factor that regulates bone marrow neutrophil retention and release[32]. No difference was observed in the expression of SDF-1 protein upon sepsis (12 hours following CS − induced sepsis) in the bone marrow of *Del1*−/− neonate mice compared to WT mice (Supplementary Fig. 8). Moreover, neutrophil numbers were significantly lower over time in *Del1*−/− mice, both in the bone marrow and in the periphery at the same time point (Figs. 4g, 6c), indicating that the effect of DEL-1

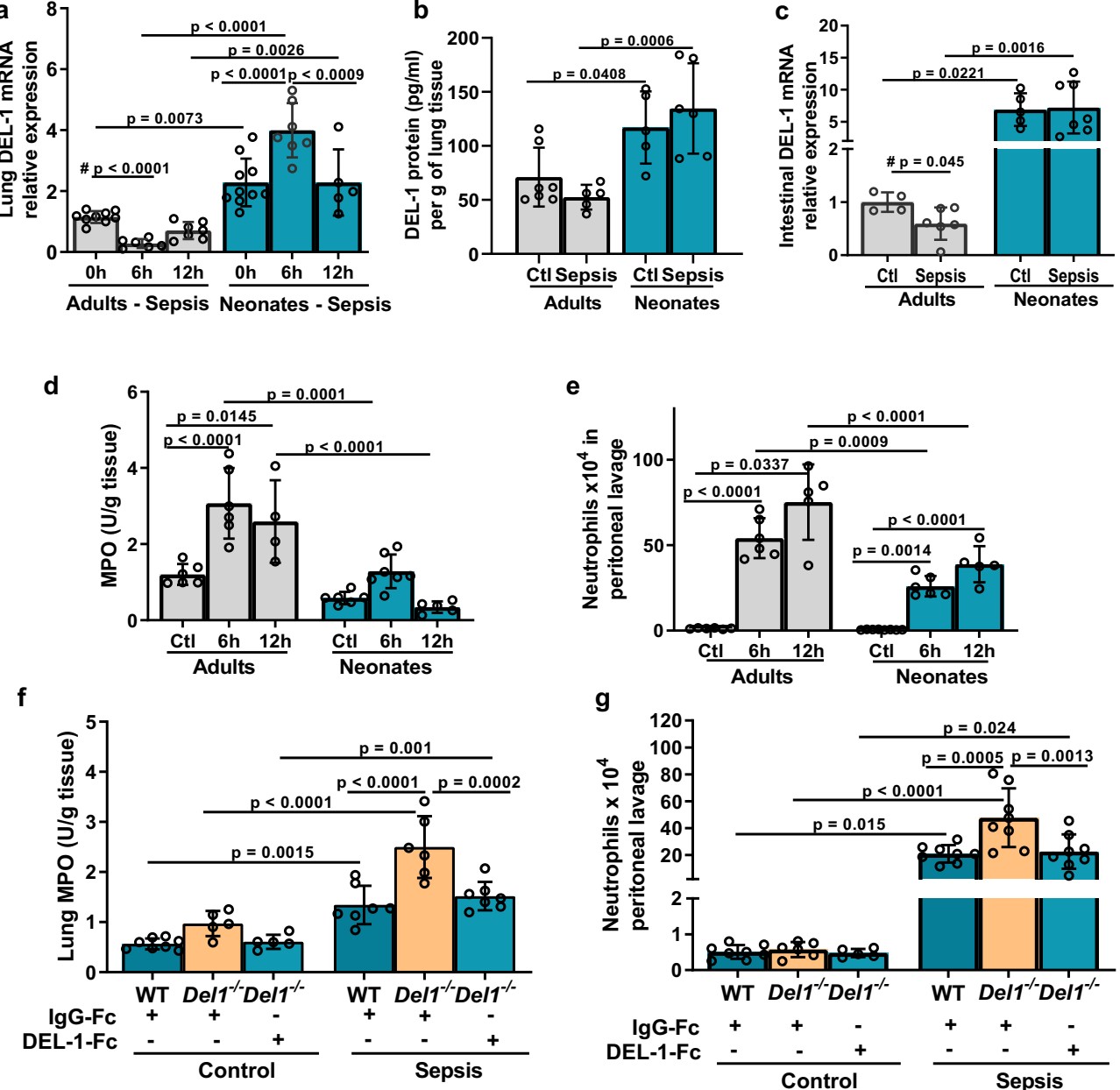

**Fig. 3 | Tissue DEL-1 expression is not suppressed upon sepsis in neonates and controls neonatal tissue neutrophil infiltration. a** DEL-1 relative mRNA expression in lung tissue from C57BL/6 mice of adult (8–10 weeks) or neonatal age (4 days old) upon polymicrobial sepsis [6 and 12 hours after cecal slurry (CS) administration] (expression of DEL-1 in adult control lungs was set as 1) (*n* = 9 animals in adult 0 h, *n* = 6 animals in adult 6 h, *n* = 7 animals in adult 12 h, *n* = 10 animals in neonates 0 h, *n* = 7 animals in neonates 6 h and *n* = 5 animals in neonates 12 h group). **b** DEL-1 protein levels in lung lysates from C57BL/6 mice of adult (8–10 weeks) or neonatal age (4 days old) following 12 hours of CS-induced polymicrobial sepsis (*n* = 7 animals in adult control, *n* = 5 animals in adult sepsis and neonate control groups and *n* = 6 animals in neonatal sepsis group). **c** DEL-1 relative mRNA expression in small intestine from C57BL/6 mice of adult (8–10 weeks) or neonatal age (4 days old) following 6 hours of CS-induced polymicrobial sepsis (expression of DEL-1 in adult control intestines was set as 1) (*n* = 4 animals in adult control group, *n* = 6 animals in adult sepsis group, *n* = 5 animals in neonate control group and *n* = 6 animals in neonatal sepsis group). **d** Myeloperoxidase (MPO) activity in total lung lysates in wild type (WT) C57BL/6 adult and neonate mice 6 and 12 hours after CS-induced sepsis (n = 6 animals in adult 0 h, *n* = 6 animals in adult 6 h, *n* = 4 animals in adult

12 h, *n* = 6 animals in neonates 0 h, *n* = 7 animals in neonates 6 h and *n* = 5 animals in neonates 12 h group). **e** Neutrophil numbers (CD11b⁺ Ly6G⁺) in the peritoneum in WT C57BL/6 adult and neonate mice 6 and 12 hours after CS-induced sepsis (*n* = 6 animals in adult 0 h, *n* = 6 animals in adult 6 h, *n* = 5 animals in adult 12 h, n = 8 animals in neonates 0 h, *n* = 6 animals in neonates 6 h and *n* = 5 animals in neonates 12 h group). **f** MPO activity in lung (*n* = 8 animals in WT +IgG-Fc control, n = 5 animals in *Del1⁻/⁻* + IgG-Fc control, *n* = 5 animals in *Del1⁻/⁻* + DEL-1-Fc control, *n* = 7 animals in WT + IgG-Fc sepsis, *n* = 6 animals in *Del1⁻/⁻* + IgG-Fc sepsis and *n* = 5 animals in *Del1⁻/⁻* + DEL-1-Fc sepsis group). and **g** total neutrophil count (CD11b⁺ Ly6G⁺) in peritoneal lavage in WT and *Del1⁻/⁻* C57BL/6 neonate mice that received either DEL-1-Fc or IgG-Fc i.v. 15 min prior to CS-induced sepsis (6 hours) (*n* = 8 animals in WT +IgG-Fc control, *n* = 6 animals in *Del1⁻/⁻* + IgG-Fc control, *n* = 5 animals in *Del1⁻/⁻* + DEL-1-Fc control and *n* = 8 animals per group in WT + IgG-Fc sepsis, *Del1⁻/⁻* + IgG-Fc sepsis and *Del1⁻/⁻* + DEL-1-Fc sepsis groups). Mean ± SD (**a**–**d**) is depicted. Statistical analysis by one-way ANOVA with Bonferroni's multiple comparison post-test (**a**–**g**), and two-sided unpaired *t* test (**a** and **c**, between adult mice indicated with #). Source data are provided as a Source Data file. Ctl control, h hours, U units, g grams.

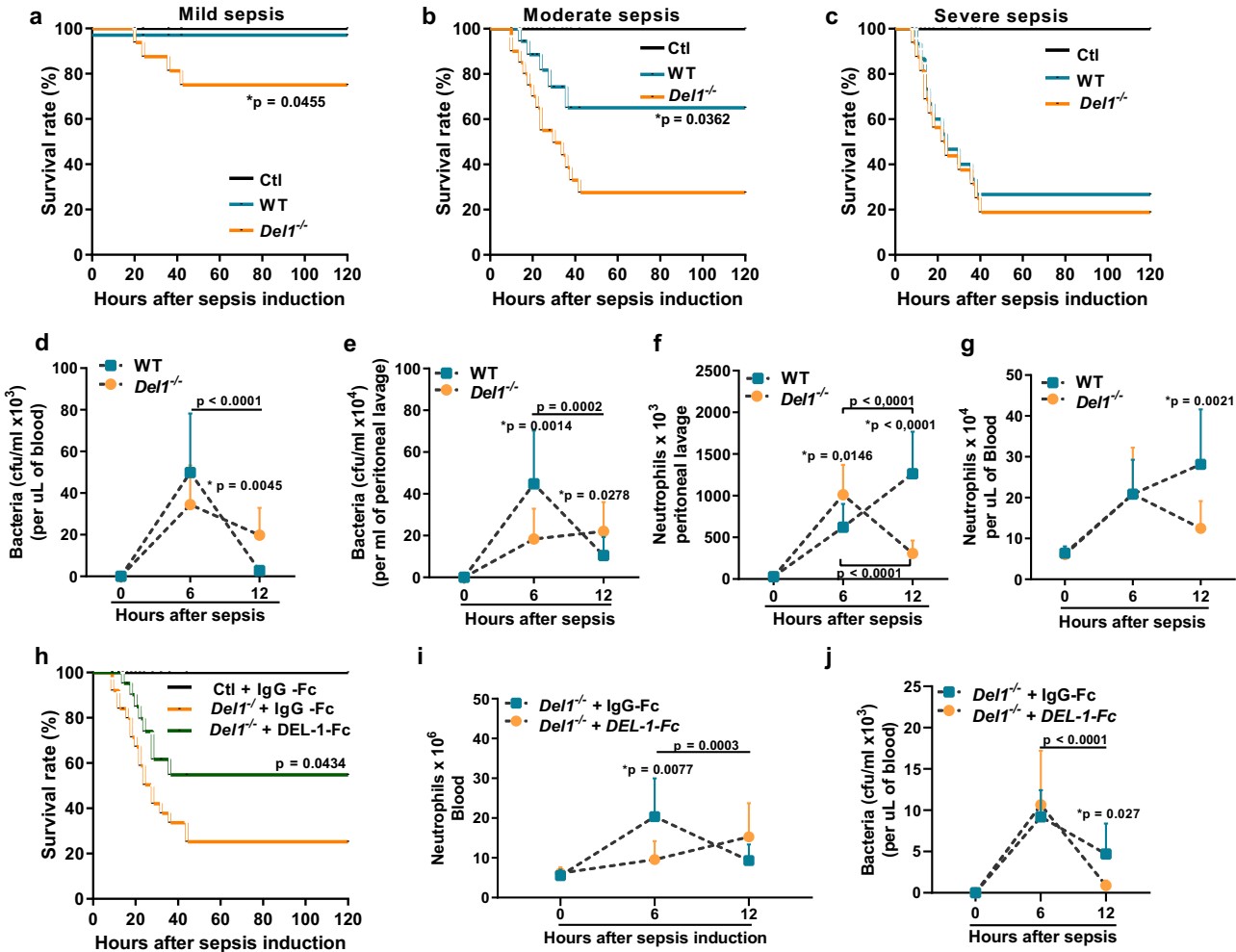

**Fig. 4 | DEL-1 promotes neonatal sepsis survival via enhancement of circulating neutrophil pool.** Survival rates of wild type (WT) and *Del1*[−/−] C57BL/6 neonate mice after cecal slurry (CS) induced sepsis of either **a** mild (n = 16 animals per group), **b** moderate (n = 16 animals per group), or **c** severe severity (n = 16 animals per group). Bacterial counts, expressed as colony forming units (cfu) in **d** blood (n = 13 animals in WT 6 h, n = 12 animals in WT 12 h, n = 14 animals in *Del1*[−/−] 6 h and n = 9 animals in *Del1*[−/−] mice 12 h group) (in the 0 h time point the value was set to 0), and **e** peritoneum (n = 13 animals in WT 6 h, n = 8 animals in WT 12 h, n = 13 animals in *Del1*[−/−] 6 h and n = 8 animals in *Del1*[−/−] 12 h) (in the 0 h time point the value was set to 0). Neutrophils (CD11b[+] Ly6G[+]) in (**f**) peritoneal lavage (n = 7 animals in WT 0 h, n = 14 animals in WT 6 h and n = 12 animals in WT 12 h, n = 6 animals in *Del1*[−/−] 0 h, n = 17 animals in *Del1*[−/−] 6 h and n = 10 animals in *Del1*[−/−] 12 h) and **g** blood (n = 4 animals in WT 0 h, n = 21 animals in WT 6 h, n = 12 animals in WT 12 h, n = 4 animals in *Del1*[−/−] 0 h, n = 19 animals in *Del1*[−/−] 6 h and n = 11 animals in *Del1*[−/−] 12 h group) in WT and *Del1*[−/−] C57BL/6 neonate mice 6 and 12 hours following CS-induced sepsis. **h** Survival rates of *Del1*[−/−] C57BL/6 neonate mice treated with i.v. DEL-1-Fc or IgG-Fc 15 min prior to injection of CS (n = 25 animals per group). **i** Neutrophils (CD11b[+]Ly6G[+]) (n = 5 animals in IgG-Fc *Del1*[−/−] 0 h, n = 17 animals in IgG-Fc *Del1*[−/−] 6 h, n = 15 animals in IgG-Fc *Del1*[−/−] 12 h, n = 5 animals in *Del1*[−/−] 0 h, n = 8 animals in DEL-1-Fc in *Del1*[−/−] 6 h and n = 10 animals in DEL-1-Fc *Del1*[−/−] 12 h group) and **j** bacterial counts (cfu) in the blood of *Del1*[−/−] neonate pups treated with either i.v. DEL-1-Fc or IgG-Fc 6 and 12 hours following CS-induced sepsis (n = 4 animals in IgG-Fc *Del1*[−/−] 6 h, n = 8 animals in IgG-Fc *Del1*[−/−] 12 h, n = 7 animals in DEL-1-Fc *Del1*[−/−] 6 h and n = 9 animals in DEL-1-Fc *Del1*[−/−] 12 h group) (in the 0 h time point the value was set to 0). Statistical analysis by Log-rank (Mantel-Cox) test in survival experiments (**a**–**c**, **h**). Mean ± SD (**d**, **e**, **f**, **g**, **i**, **j**) is depicted. Statistical analysis by one-way ANOVA with Bonferroni's multiple comparison post-test (**d**–**g**, **i**, **j**) or by two-sided unpaired *t* test (**d**–**g**, **i**, **j**, indicated with *) to compare between WT and *Del1*[−/−] or IgG-Fc *Del1*[−/−] and DEL-1-Fc *Del1*[−/−] groups at a specific timepoint following CS-induced sepsis. Source data are provided as a Source Data file. Ctl control.

deficiency on bone marrow neutrophil numbers cannot be attributed to increased release of neutrophils from the bone marrow to the peripheral blood upon infection.

Upon systemic microbial infection, proliferation of hematopoietic progenitors in the bone marrow is activated to compensate for the increased need for mature myeloid cells and restore bone marrow cellularity, a process known as emergency myelopoiesis[33,34]. We therefore assessed whether DEL-1 affects the hematopoietic progenitor numbers in the bone marrow, both at steady state and upon septic conditions. Although DEL-1 deficiency had no significant effect on long-term HSC (LT-HSCs) and short-term HSC (ST-HSCs) population (Lin[−]cKit[+]Sca1[+]CD48[−] CD150[−]), we observed a significant decrease in the number of multipotent progenitors (MPPs, Lin[−]Sca-

1[+]cKit[+]CD48[+]CD150[−]) and particularly the myeloid-biased MPP3 lineage (MPP3s, Lin[−]Sca-1[+]cKit[+]Flt3[−]CD48[+]CD150[−]) in *Del1*[−/−] mice, upon polymicrobial sepsis (Fig. 7a–d)[33,35].

The decrease of MPP3 cells was associated with a significant decrease of the percentage of granulocyte-macrophage progenitors (GMPs) (Lin[−]cKit[+]Sca1[−]CD16/32[+]CD34[+]) in the bone marrow of septic *Del1*[−/−] neonates, compared to WT neonate mice (Fig. 8a, b). In the context of sepsis, DEL-1-Fc administration significantly prevented the decline of GMPs in the bone marrow of DEL-1 deficient neonate pups (Fig. 8c), further supporting that DEL-1 promotes emergency granulopoiesis. It has been previously shown that DEL-1 induces HSC proliferation and differentiation toward the myeloid lineage under stress conditions in the bone marrow of adult mice via interaction with β3

integrin (CD61) on HSCs[23]. We observed that the DEL-1 receptor avβ3 integrin (comprising the av integrin CD51 and the β3 integrin CD61) was expressed in the bone marrow of neonate mice to a similar extent with adult mice, in either steady state or upon sepsis (assessed by flow cytometry analysis in bone marrow total cells, Supplementary Fig. 9a, b).

As G-CSF is a major regulator of steady-state and emergency myelopoiesis[34,36,37], we examined whether serum G-CSF is altered in *Del1*[-/-] neonate mice compared to WT controls. We did not observe any alterations in the expression G-CSF in serum in *Del1*[-/-] neonate mice compared to WT, upon steady state or septic conditions (Supplementary Fig. 10). Moreover, the serum expression of inflammatory cytokines involved in emergency myelopoiesis (such as TNF, IL-6, IL-17, IL-10, CXCL-1) was not different in septic *Del1*[-/-] and WT neonate mice (Supplementary Figs. 3a–d and 11a–c). Therefore, the defect in myelopoiesis that is noted in DEL-1 deficiency cannot be attributed to alterations in the expression of G-CSF or other inflammatory mediators.

Overall, these findings indicate that DEL-1 acts as a niche factor that regulates the numbers of multipotent progenitors and GMPs and, hence, neutrophil production in the bone marrow of neonates upon systemic bacterial infection.

## IL-10 expression in neonates promotes DEL-1 upregulation in sepsis

DEL-1 is negatively regulated by inflammatory cytokines such as TNF and IL-17A[24,34,36]. To identify the key cytokines that control DEL-1 expression in septic neonates, we analyzed serum expression of inflammatory cytokines (TNF, IL-17A, IL-6) and the anti-inflammatory cytokine IL-10 in septic neonate and adult mice. Expression of TNF in septic neonates was similar to adults, IL-17A expression was less in septic neonates, while IL-6 and IL-10 were higher in septic neonates compared to adults (Fig. 9a–d). Unlike TNF and IL-17A that are known to downregulate DEL-1[24], the effect of IL-6 and IL-10 on DEL-1 expression in endothelial cells has not been previously examined. To this end, we treated human endothelial cells (human umbilical vein endothelial

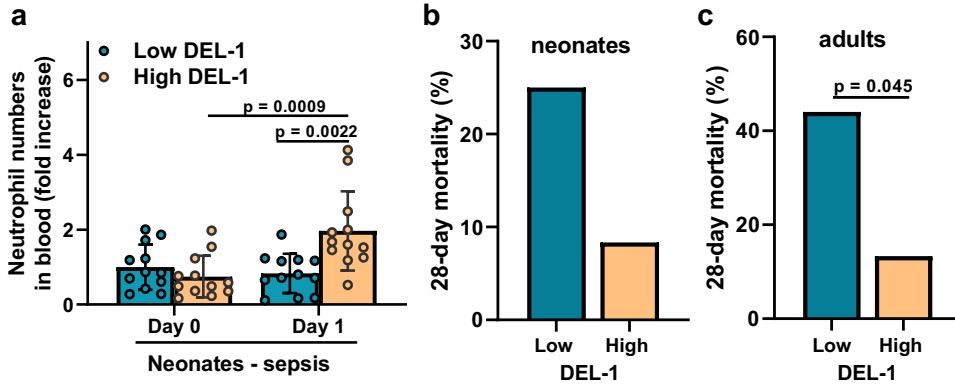

**Fig. 5 | DEL-1 expression correlates with enhanced circulating neutrophils and lower mortality in septic human subjects. a** Fold increase in neutrophil numbers within 24 hours in human neonates with sepsis with either low serum DEL-1 concentration (<700 pg/ml) or high serum DEL-1 concentration (>700 pg/ml) (in the group with low serum DEL-1, at day 0, the neutrophil numbers were set as 1) (*n* = 12 neonates per group). 28-day mortality rate (%) in (**b**) septic neonates (*n* = 12 neonates per group) and (**c**) adults (*n* = 25 adults in the group wtih low serum DEL-1

concentration and *n* = 15 adults in the group with high serum DEL-1 concentration) with either low DEL-1 or high DEL-1 concentration in the serum (the threshold for high/low DEL-1 concentration was determined to 125 pg/ml for adult patients and 700 pg/ml for neonates). Mean ± SD (**a**) and frequency (%) (**b**, **c**) are depicted. Statistical analysis with one-way ANOVA with Bonferroni multiple comparison post-test (**a**) and Chi-squared test (for 28-day mortality categorical data **b**, **c**). Source data are provided as a Source Data file.

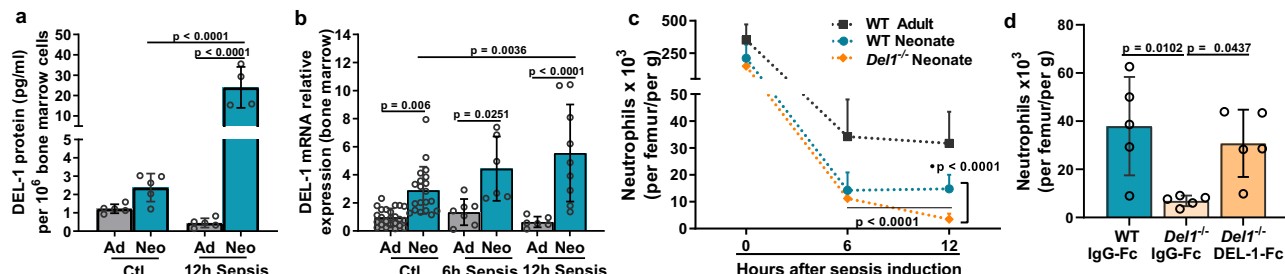

**Fig. 6 | DEL-1 supports bone marrow neutrophil pool in neonatal sepsis. a** DEL-1 protein expression (pg/ml) per 10⁶ bone marrow cells from wild type (WT) C57BL/6 mice of adult and neonatal age at basal state and following 12 hours of cecal slurry (CS) -induced sepsis (*n* = 5 animals per group in adult control, neonate control, and adult sepsis group and *n* = 4 animals in neonate sepsis group). **b** DEL-1 mRNA relative expression in total bone marrow cells from WT C57BL/6 mice of adult and neonatal age at basal state and following 6 and 12 hours of CS-induced sepsis (expression of DEL-1 in adult control mice was set as 1) (*n* = 24 animals in adult control, *n* = 20 animals in neonate control, *n* = 7 animals in adult 6 h sepsis, *n* = 6 animals in neonate 6 h sepsis, *n* = 6 animals in adult 12 h sepsis and *n* = 9 animals in neonate 12 h sepsis group). **c** Total neutrophil numbers (CD11b⁺Ly6G⁺Ly6C⁻, normalized per femur and per mouse weight in grams) in bone marrow of WT adult, WT and *Del1*[-/-] C57BL/6

neonate pups in steady state and following 6 and 12 hours of CS induced sepsis (*n* = 7 animals in WT adult control group, *n* = 5 animals per group in WT adult 6 h and 12 h sepsis, *n* = 16 animals in WT neonate control, *n* = 10 animals per group in WT neonate 6 h and 12 h sepsis, *n* = 7 animals in *Del1*[-/-] control, *n* = 9 animals in *Del1*[-/-] 6 h sepsis and *n* = 10 animals in *Del1*[-/-] 12 h sepsis). **d** Total neutrophil numbers (CD11b⁺Ly6G⁺Ly6C⁻, normalized per femur and per mouse weight in grams) in bone marrow of WT neonates, and *Del1*[-/-] neonate pups treated with i.v. DEL-1-Fc or IgG-Fc 15 min prior to injection of CS and following 12 hours of CS-induced sepsis (*n* = 5 animals per group). Mean ± SD is depicted (**a–d**). Statistical analysis by one-way ANOVA with Bonferroni's and multiple comparison post-test (**a**, **b**, **d**) and two-sided unpaired *t* test between the indicated groups (**c**). Source data are provided as a Source Data file. Ctl control, Ad adults, Neo neonate, h hours, g grams.

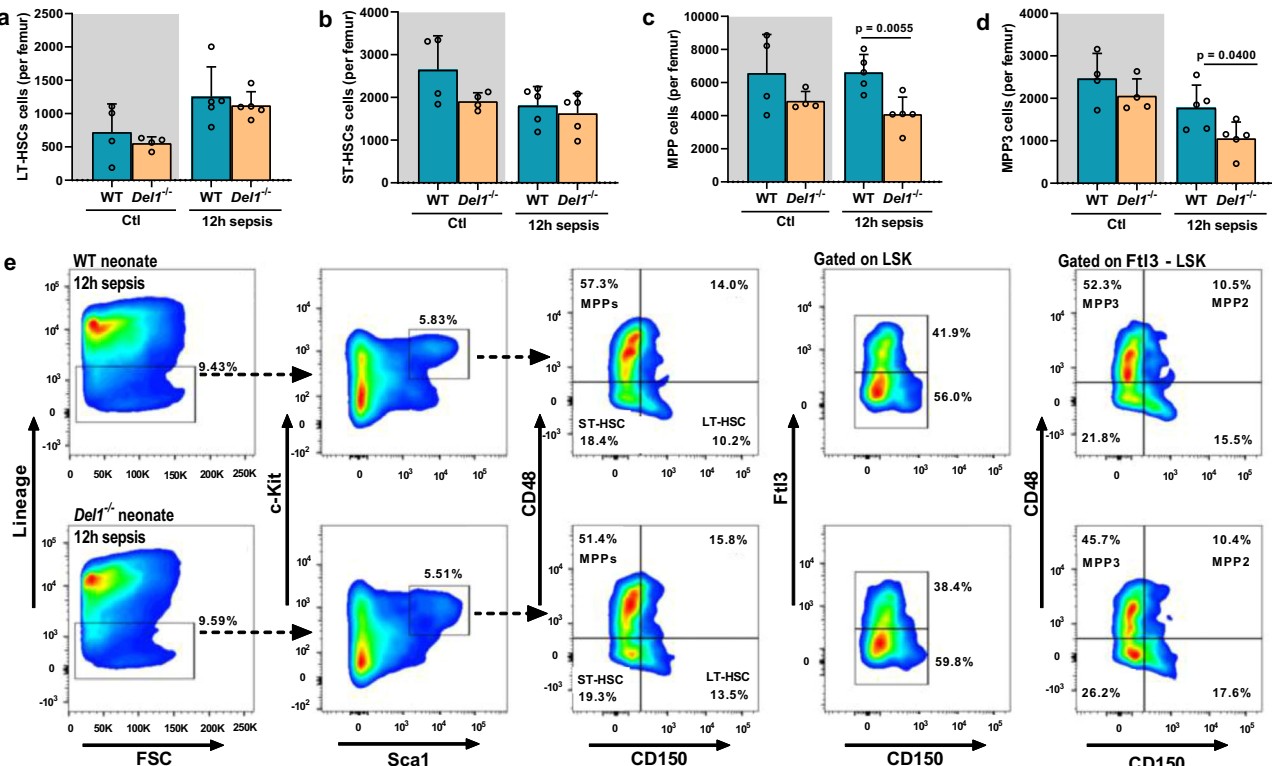

**Fig. 7 | DEL-1 supports multipotent progenitors upon sepsis in neonates.**
**a** Long-term hematopoietic stem cells (LT-HSCs, CD48⁻CD150⁺LSK) [$n = 4$ animals per group in control wild type (WT) and $Del1^{-/-}$ groups, and $n = 5$ animals per group in septic WT and $Del1^{-/-}$ groups] **b** short-term HSCs (ST-HSCs, CD48⁻CD150⁻LSK) ($n = 4$ animals per group in control WT and $Del1^{-/-}$ groups, and $n = 5$ animals per group in septic WT and $Del1^{-/-}$ groups) **c** multipotent progenitors (MPPs, CD48⁺CD150⁻LSK) and ($n = 4$ animals per group in control WT and $Del1^{-/-}$ groups, and $n = 5$ animals per group in septic WT and $Del1^{-/-}$ groups) **d** myeloid-biased MPPs (MPP3s, Flt3⁻CD48⁺CD150⁻LSK) ($n = 4$ animals per group in control WT and $Del1^{-/-}$ control groups and $n = 5$ animals per group in septic WT and $Del1^{-/-}$ groups) and **e** representative flow cytometry plots of MPP and MPP3 cells in the bone marrow of WT and $Del1^{-/-}$ C57BL/6 neonate mice at steady state (gray background) and at 12 hours of cecal slurry–induced polymicrobial sepsis. Mean ± SD is depicted (**a–d**). Statistical analysis by unpaired two-sided $t$ test between WT and $Del1^{-/-}$ control groups (gray background, **a–d**) and between WT and $Del1^{-/-}$ septic groups (**a–d**). Source data are provided as a Source Data file. LSK (Lin⁻Sca-1⁺cKit⁺), Ctl control, h hours.

cells, HUVECs and the Ea.hy926 endothelial cell line) with human recombinant IL-6 or IL-10. IL-6 did not affect DEL-1 expression, while IL-10 promoted a two-fold upregulation of DEL-1 mRNA expression (Fig. 9e, f). Stimulation with IL-10 reversed the suppressive effect of IL-17A on DEL-1 expression (Fig. 9e, f), indicating opposing actions of IL-10 and IL-17A on DEL-1 expression.

To determine whether differences in DEL-1 expression observed in different tissues during neonatal sepsis reflected respective changes in IL-17A and IL-10, we measured IL-17A and IL-10 mRNA expression in tissue extracts from neonates 6 hours following exposure to CS-induced sepsis. We found that in neonatal lung, intestine and kidney, where DEL-1 was not suppressed in CS sepsis (Fig. 2a, b), a higher IL-10 to IL-17A ratio was observed (Fig. 9g). Consistent with these findings in mouse tissues, the median human serum DEL-1 protein concentration was higher in septic adults and septic neonates that exhibited high (>2) serum IL-10 to IL-17A ratio, compared to those with low (<2) IL-10 to IL-17A ratio (Fig. 9h), further supporting that the IL-10/IL-17A balance regulates DEL-1 levels in sepsis. Moreover, we showed that IL-10 induced DEL-1 expression in human mesenchymal stromal cells (MSCs) (Fig. 9i), suggesting that this cytokine might control DEL-1 expression in the human bone marrow.

To determine whether IL-10 regulates DEL-1 expression in the bone marrow during sepsis, we treated neonate septic mice with an IL-10 receptor-blocking antibody (anti-IL-10R). Administration of anti-IL-10R led to suppression of DEL-1 mRNA expression in the bone marrow of WT neonates (Fig. 9j) and reduction of the neutrophil pool in the bone marrow and blood (Fig. 9k, left panel). In stark contrast, anti-IL-10R failed to affect the bone marrow neutrophil pool in $Del1^{-/-}$ mice,

which exhibited similar neutrophil numbers regardless of whether they were treated with anti-IL-10R or IgG (Fig. 9k, right panel). Importantly, treatment of WT neonate septic mice with anti-IL-10R resulted in higher bacterial load in blood (Fig. 9l) and reduced survival (Fig. 9m). Together, these findings indicate that the IL-10/DEL-1 axis promotes neutrophil production and host survival under septic stress conditions in neonates (Fig. 10).

## Discussion

In this study, we demonstrate that the endogenous soluble molecule DEL-1 is elevated in early life under normal and septic conditions in both murine and human neonates. DEL-1 expression is under the control of IL-10 and is essential to promote emergency granulopoiesis in neonatal sepsis, thereby facilitating sustained output of circulating neutrophils, control of bacteremia and survival from sepsis. Unlike earlier studies where the protective effect of DEL-1 was dependent predominantly on its anti-inflammatory/anti-leukocyte recruitment and pro-resolution properties[18,20–22], in the present study the mechanism whereby DEL-1 protects mice from sepsis appears to depend strongly on its ability to support stress granulopoiesis. Additionally, in this study we identify IL-10 as a positive regulator of DEL-1 production. Although other endogenous molecules, such as specialized pro-resolution mediators (D-resolvins) were shown earlier to promote DEL-1 expression during inflammation resolution[38], IL-10 is unique in that it can induce DEL-1 also under inflammatory conditions. Hence, IL-10 has an important role in sustaining DEL-1 production under sepsis.

Tissue $Del1$ mRNA expression and serum DEL-1 protein expression were elevated in the neonatal period in mice and human neonates

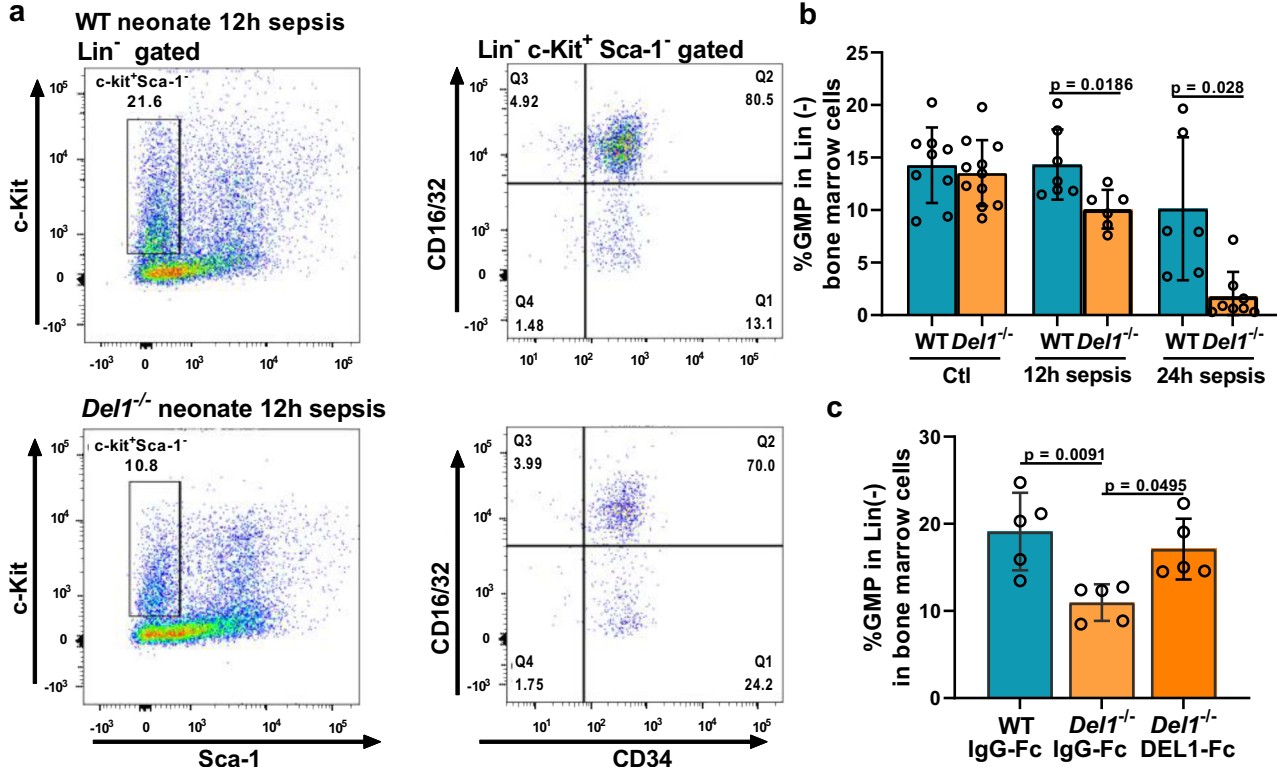

**Fig. 8 | DEL-1 prevents the decline of granulocyte progenitors upon sepsis in neonates. a** Representative flow cytometry plots of granulocyte-macrophage progenitors (GMPs) (Lin⁻c-Kit⁺ Sca-1⁻CD16/32⁺CD34⁺) in wild-type (WT) and *Del1⁻/⁻* C57BL/6 neonate pups at 12 hours of cecal slurry (CS)-induced sepsis **b** percentage of GMPs in WT and *Del1⁻/⁻* C57BL/6 neonate pups at basal state and following 12 and 24 hours of CS-induced sepsis (*n* = 9 animals in control WT, *n* = 11 animals in *Del1⁻/⁻* control, *n* = 7 animals in WT 12 h sepsis, *n* = 6 animals in *Del1⁻/⁻* 12 h sepsis, *n* = 6 animals in WT 24 h sepsis and *n* = 8 animals in *Del1⁻/⁻* 12 h sepsis groups). **c** percentage of GMPs in WT neonate mice and *Del1⁻/⁻* C57BL/6 neonate mice treated with i.v. DEL-1-Fc or IgG-Fc 15 min prior to injection of CS and following 12 hours of CS-induced sepsis (*n* = 5 animals per group). Mean ± SD is depicted (**b, c**). Statistical analysis by one-way ANOVA with Bonferroni's multiple comparison post-test (**c**) and unpaired two-sided *t* test between WT and *Del1⁻/⁻* control groups of the same time point (**b**, as indicated). Source data are provided as a Source Data file. Ctl control, h hours.

respectively, and gradually reached levels similar to older subjects as postnatal age advanced. Neonate mice expressed higher levels of DEL-1 compared to adult mice in all tissues studied apart from brain. Notably, DEL-1 expression was detected at tissue level only, as circulating DEL-1 was not detected in the murine model. This finding is in accordance with published literature showing that DEL-1 is acting mostly locally, being a regulator of local tissue inflammation with its functions depending on the location of its expression[13,38].

The differential expression of DEL-1 in tissues observed here is in accordance with a previous report in adult mouse tissues, indicating that the tissue-related requirements for immune privilege are unaffected by age[14]. Contrary to what was observed in adults, DEL-1 was not downregulated upon sepsis in neonates. DEL-1 expression is transiently suppressed locally in tissues in response to pro-inflammatory triggers in adult mouse models of LPS-induced lung injury, cecal ligation and puncture (CLP)-induced sepsis, inflammatory bone loss, experimental autoimmune encephalomyelitis and sepsis-induced adrenal gland inflammation[8,14,20,21,25,39]. Our results demonstrate that in septic neonate mice, DEL-1 expression does not follow the same expression pattern as in adult mice, which was attributed to the higher IL-10 to IL-17A ratio in neonates compared to adults.

The role of DEL-1 in the outcome of sepsis in neonates has not been previously evaluated. Our findings showed that DEL-1 deficiency had a substantial negative impact on survival of septic murine neonates. Furthermore, low serum DEL-1 concentration in adult and neonate patients with sepsis were correlated with reduced 28-day survival compared to septic adult and neonate individuals with high serum

DEL-1. DEL-1 has been reported to promote homeostasis and improve the outcome of disease in adult experimental models of periodontal inflammation, arthritis, experimental autoimmune encephalomyelitis and LPS-induced adrenal gland inflammation[8,14,20,21,40,41]. On the contrary, in a study in CLP-induced sepsis in adult mice and in adult human septic patients, high DEL-1 protein levels in the serum were correlated with increased mortality[42]. This discrepancy supports the complexity of sepsis pathogenesis and may also be attributed to the differences in immune responses and bone marrow reserves between adults and neonates as well as to differential kinetics of tissue DEL-1 expression compared to serum DEL-1.

Neonatal leukocytes exhibit reduced ability to adhere and transmigrate via the endothelium to the tissues during inflammatory stimuli[43–45] and based on our results, the increased DEL-1 concentration in the neonatal period could at least contribute to this phenomenon in mice. It has been suggested though, that insufficient leukocyte recruitment in fetal life and in the newborn period may be one of the reasons for the high incidence of severe infections[44,46,47]. Based on our findings, increased tissue leukocyte infiltration in murine neonates was negatively associated with sepsis survival, as *Del1⁻/⁻* mice had higher neutrophilic infiltration in tissues compared to WT ones, but worse survival and, consistently, DEL-1 administration abrogated the exaggerated tissue neutrophil infiltration but improved survival from sepsis. In accordance with our observation, in a previous study in adult murine model of polymicrobial sepsis, ICAM-1-deficient mice demonstrated a significant reduction of mortality compared to WT mice[48]. Therefore, although leukocyte tissue recruitment is essential for pathogen elimination, it may provoke an overwhelming inflammatory

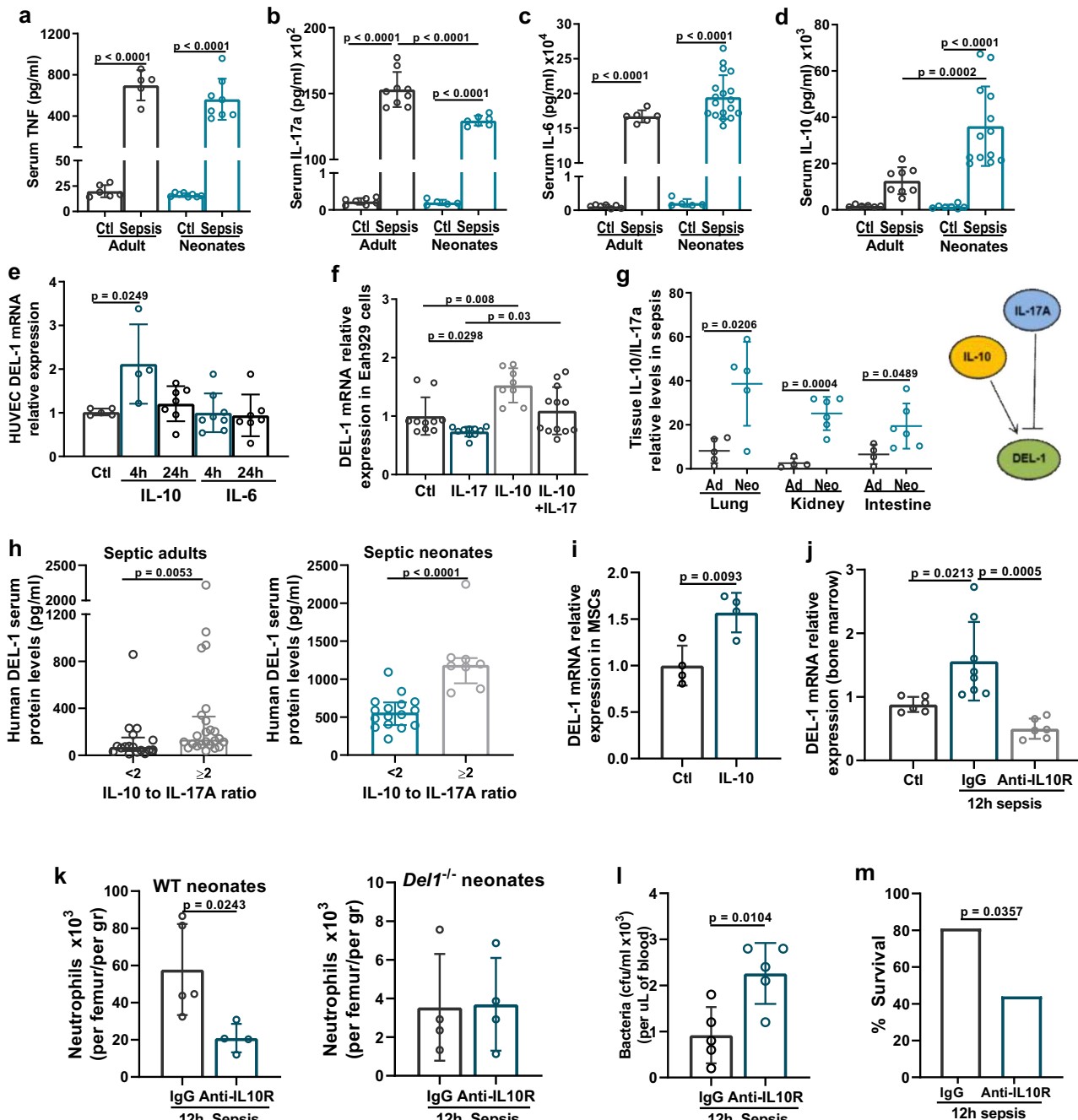

response that may lead to microvasculature injury and contribute to organ damage and failure[9,49–54]. Our finding that enhanced leukocyte recruitment was detrimental in septic *Del1⁻/⁻* neonates, highlights the impact of DEL-1 in controlling exaggerated organ inflammation and damage in young hosts.

Previous reports have shown that DEL-1 attenuates complement-dependent phagocytosis of red blood cells through inhibition of complement receptor 3 (Mac-1-integrin)[55]. However, neonates are partially deficient in opsonins, such as complement and IgG, and non-opsonic phagocytosis is prominent compared to adults[29–31]. In this study, we investigated phagocytosis with non-opsonised bacteria, to recapitulate the phagocytosis conditions in early life. Under these conditions, we found that DEL-1-Fc did not affect the phagocytosis capacity of neonatal neutrophils in mice, suggesting that the higher susceptibility to sepsis in neonate *Del1⁻/⁻* mice was not linked to altered phagocytosis.

Our report expands our knowledge by demonstrating that DEL-1 is essential to maintain emergency granulopoiesis in the bone marrow of neonate mice and is required to prevent sepsis-induced neutropenia. In the bone marrow, there is a pool of neutrophil precursors that maintain the ability to multiply and rapidly replenish neutrophil numbers in the circulation upon sepsis[56,57]. In term neonates, the bone marrow neutrophil pool is greatly diminished compared to adults and, moreover, have a limited ability to recruit or generate new neutrophils during infections, contributing to the development of neutropenia[58,59]. In accordance to the above, WT neonate mice in our study had significantly lower numbers of neutrophils in the bone marrow at steady-state and in sepsis, compared to adults. Interestingly, neutrophil numbers were significantly decreased over time in *Del1⁻/⁻* neonate mice compared to WT ones, both in the bone marrow and in the periphery as sepsis progressed. In addition, SDF-1, the bone marrow niche factor that regulates

**Fig. 9 | IL-10 promotes DEL-1 expression in neonatal sepsis.** Protein expression of (**a**) TNF ($n = 6$ animals in adult control, $n = 5$ animals in adult sepsis, $n = 8$ animals in neonate control and $n = 8$ animals in neonate sepsis group), **b** IL-17A ($n = 7$ animals in adult control, $n = 9$ animals in adult sepsis, $n = 5$ animas in neonate control and $n = 7$ animals in neonate sepsis group), **c** IL-6 ($n = 7$ animals in adult control, $n = 6$ animals in adult sepsis, $n = 5$ animals in neonate control and $n = 18$ animals in neonate sepsis group) and **d** IL-10 ($n = 6$ in adult control, n = 8 in adult sepsis, $n = 7$ in neonate control and $n = 13$ in neonate sepsis group) protein expression in serum of C57BL/6 neonate and adult mice 6 hours after exposure to cecal slurry (CS)−induced sepsis. **e** DEL-1 mRNA relative expression in HUVECs at several time points upon stimulation with human recombinant IL-10 or IL-6 protein (expression of DEL-1 in control group was set as 1) ($n = 5$ biological replicates in control, $n = 4$ biological replicates in IL-10 4 h, $n = 6$ biological replicates in IL-10 24 h, $n = 8$ biological replicates in IL-6 4 h group and $n = 7$ biological replicates in IL-6 24 h group). **f** DEL-1 mRNA relative expression in Ea.hy926 endothelial cells upon stimulation with human recombinant IL-17, IL-10 or both, for 4 hours (expression of DEL-1 in control group was set as 1) ($n = 10$ biological replicates in control, $n = 10$ biological replicates in IL-10, $n = 8$ biological replicates in IL-17 and $n = 12$ biological replicates in IL-10 and IL-17 group). **g** Tissue (lung, kidney and intestine) IL-10 to IL-17A mRNA ratio in C57BL/6 neonate and adult mice exposed to CS sepsis (6 hours) ($n = 5$ animals per group in adult and neonatal lung groups, $n = 4$ animals in adult and $n = 6$ animals in neonatal kidney groups, $n = 4$ animals in adult and $n = 6$ animals in neonatal intestine groups and $n = 4$ animals per group on adult and neonatal brain tissue groups). **h** Median human DEL-1 protein in the serum of septic adults and neonates that exhibited either low (<2) or high (≥2) serum IL-10 to IL-17 ratio ($n = 17$ adults in ≤2 group, $n = 24$ adults in ≥2 group, $n = 16$ neonates in ≤2 group, and $n = 8$ neonates ≥2 group). **i** DEL-1 mRNA relative expression in human mesenchymal stromal cells upon 4 hours of IL-10 stimulation (expression of DEL-1 in control group was set as 1) ($n = 4$ biological replicates per group). **j** DEL-1 mRNA expression in murine bone marrow of wild type (WT) neonate mice that received either i.v anti-IL-10R (anti-IL-10 receptor) or IgG (expression of DEL-1 in control group was set as 1) and subjected to CS-induced sepsis (12 hours) ($n = 6$ animals in control, $n = 8$ animals in septic IgG group and $n = 6$ animals in septic a-IL-10R group). **k** Total neutrophils in bone marrow of WT neonate mice (left panel) and *DelI$^{-/-}$* neonate mice (right panel) that received either i.v. anti-IL-10R or IgG and subjected to CS-induced sepsis (12 hours) ($n = 5$ animals in IgG WT, $n = 4$ animals in a-IL-10R WT n = 4 animals in IgG *DelI$^{-/-}$* group and $n = 4$ animals in septic a-IL-10R *DelI$^{-/-}$* group), **l** Bacterial colony forming units (cfu) in the blood of WT neonate septic mice that received i.v. either anti-IL-10R or IgG ($n = 5$ animals in septic IgG group and $n = 5$ in septic anti-IL-10R group) and **m**, survival in WT neonate septic mice that received i.v. either anti-IL-10R or IgG and subjected to CS-induced sepsis ($n = 21$ animals in septic IgG group and $n = 16$ animals in septic a-IL-10R group). Mean ± SD (**a**–**g**, **i**–**k**), median ± interquartile range (**h**) and frequency % (**m**) are depicted. Statistical analysis by one-way ANOVA with Bonferroni's multiple comparison post-test (**a**–**e**, **j**), two-sided Mann-Whitney test (h) unpaired two-sided *t* test (**f, g**, i, **k, l**) and Fisher's exact test (**m**). Source data are provided as a Source Data file. Ctl control, Ad adults, Neo neonates, h hours.

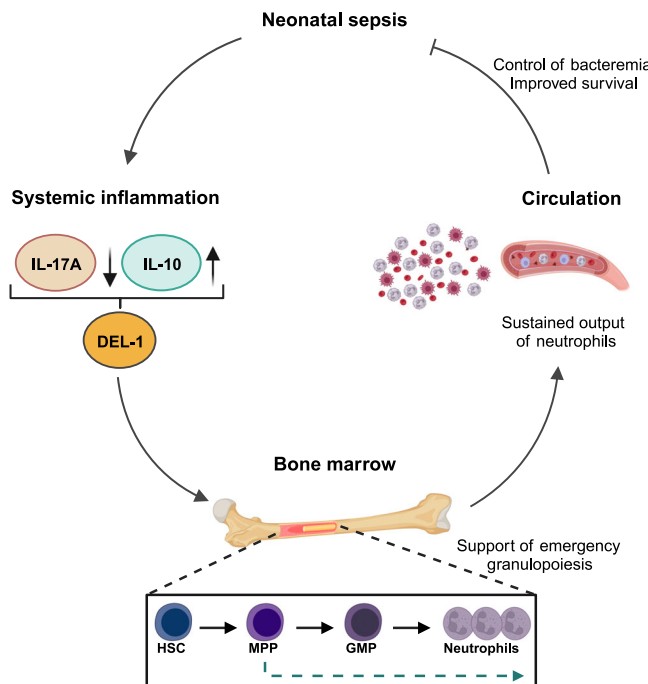

**Fig. 10 | Mechanisms of IL-10/DEL-1 mediated protection in sepsis in early life.** Schematic illustration of the main findings of this study. Upon neonatal sepsis, a distinct cytokine response is developed, characterized by significant expression of interleukin (IL)-10 and reduced expression of IL-17 compared to adult sepsis. High IL-10 to IL-17 ratio augments DEL-1 expression in sepsis in early life. DEL-1 subsequently promotes emergency granulopoiesis in septic neonate mice, via support of myeloid-biased (MPPs) and granulocyte macrophage progenitors (GMPs) in the bone marrow. Emergency granulopoiesis further supports neutrophil production in the bone marrow and sustains the output of neutrophils in the peripheral circulation, leading to control of bacteremia and survival from sepsis in early life. Created with BioRender.com.

Previous research has shown that the generation of new granulocytes from the bone marrow is defective and delayed in neonates compared to adults following CS-induced sepsis, but the mechanism underlying this phenomenon remains unclear[12]. It has been recently shown that DEL-1, derived from endothelial and mesenchymal stromal cells in the bone marrow niche, promotes myelopoiesis predominantly under hematopoietic stress conditions in a juxtracrine manner, via integrin-dependent interactions with hematopoietic progenitors[13,23]. In agreement with these findings, we show that *DelI$^{-/-}$* neonate mice exhibited excessive depletion of the bone marrow multipotent progenitors. We found that DEL-1 deficiency did not have any effect in LT-HSC or ST-HSC numbers under neither septic nor steady state conditions, which is in accordance with previous studies showing that niche factors do not necessarily have a quantitative effect on those populations[23,60,61]. Interestingly, however, DEL-1 deficiency resulted in a significant decrease in multipotent and myeloid-biased progenitors in the neonatal bone marrow under septic conditions, which affected mostly neutrophils without influencing other populations, such as monocytes in the bone marrow of neonates. The above findings were independent of G-CSF or other inflammatory cytokines. We also found that DEL-1 is elevated in the bone marrow of neonates compared to adults and was significantly induced in sepsis. These observations overall indicate that DEL-1 acts as a niche factor that supports myeloid-biased progenitors in the bone marrow upon systemic microbial infection in early life. These findings are in accordance to a previous study in an adult murine model of LPS-induced inflammation showing that DEL-1 deficiency compromised the ability of the bone marrow niche microenvironment to support proliferation and differentiation of LT-HSCs into the myeloid lineage[23]. In our study, the effect of DEL-1 on granulopoiesis progenitors was observed only in septic mice, pointing out that the DEL-1 supports granulopoiesis in early life during systemic microbial infection.

The cellular source of DEL-1 in the bone marrow has been shown to be niche cells that have a major role in the maintenance of HSCs (i.e., arteriolar endothelial cells, osteoblastic lineage cells and perivascular mesenchymal stromal cells)[13,23]. It is known that the mesenchymal stromal cell pool is expanded in early life[62]. Therefore, the increased numbers of mesenchymal stromal cells in the bone marrow niche of neonates, which also upregulate DEL-1 in response to IL-10 as presented here, may be the main source of DEL-1 in the neonatal bone marrow.

neutrophil retention, was not affected in the absence of DEL-1. Therefore, the reduced number of neutrophils in the bone marrow of *DelI$^{-/-}$* mice cannot be attributed to their increased release from the bone marrow to peripheral blood.

We further explain the mechanistic basis of the discrepancy in DEL-1 expression between neonates and adults in sepsis, by demonstrating that DEL-1 is under the control of IL-10 and particularly of the IL-10 to IL-17A ratio. IL-10 administration promoted DEL-1 expression in both endothelial and mesenchymal stromal cells, two major cell sources of DEL-1. In accordance with our findings, IL-10 production in response to pro-inflammatory stimuli in neonates has been shown to inhibit neutrophil migration to tissues[63,64]. Additionally, IL-10 administration has been shown to promote survival in a neonatal mouse model of sepsis or neonatal group B streptococcal infection[65,66].

TNF and IL-17A upregulation, as well as bacterial products, have been previously shown to suppress DEL-1 expression[8,20,24]. In accordance to this finding, we showed that not only IL-10, but the balance of IL-10 and the pro-inflammatory cytokine IL-17A, determined the expression of DEL-1 in murine tissues and in serum of septic patients, regardless of age. Finally, DEL-1 suppression in the bone marrow of WT septic mice treated with anti-IL-10R antibody also led to diminished neutrophil pool in both the bone marrow and the circulation, and had a negative impact on survival from sepsis, further supporting the role of the IL-10/DEL-1 axis in the regulation of emergency granulopoiesis and sepsis outcome in neonates.

Based on our results, IL-10-induced DEL-1 is central in promoting emergency granulopoiesis, maintenance of neutrophilia and improving host survival in neonate mice suffering from polymicrobial sepsis. This mechanism may not be crucial in adult sepsis where the bone marrow reserves are adequate, as shown in the present study. Neutropenia in neonates is a known risk factor for bacterial infection and has been linked with adverse outcome in sepsis[67]. Together, these data strongly suggest that DEL-1 remains elevated in the neonatal period to compensate for the small size of the neutrophil pool at this young age. Ultimately, neutrophil numbers rise over the first few weeks of life to achieve adult values by 4 weeks of age[68] and by this time also DEL-1 expression decreases and reaches adult levels.

Our study has several limitations. Although the CS sepsis model is currently the preferred method of studying sepsis in neonate mice[69], it is well recognized that no single animal model can recapitulate the complex and varied clinical manifestations of the sepsis syndrome in humans[70]. The data on murine sepsis that we present in this study may not necessarily be relevant in the context of human sepsis. A great variety of confounding factors, such as physiological and environmental factors as well as differences in patient management of septic humans, cannot be controlled in murine models of sepsis. In addition, in this study, we mostly studied the expression and role of DEL-1 in term murine neonates, and we measured DEL-1 in the serum of late preterm (>34 weeks GA) or term human neonates. Term and late preterm neonates have higher mortality from sepsis compared to older children and adults; however, we should acknowledge the highest susceptibility and mortality to sepsis in early life is noted in very preterm (28–32 weeks GA) or even extremely preterm (<28 weeks GA) newborns, who were not examined in this study. Further studies are required to address the function and role of DEL-1 in human sepsis, especially in very preterm or extremely premature infants.

In conclusion, to date, there are no reliable prognostic factors for sepsis outcome in neonates and there is no effective immunomodulatory strategy to simultaneously control bone marrow neutrophil production and tissue recruitment upon infection. Trials investigating the clinical use of recombinant granulocyte-macrophage colony-stimulating factor that solely increase neutrophil numbers in preterm infants have yielded disappointing results[71,72]. The critical functions of DEL-1, its correlation with the neutrophilic response and survival from sepsis and the fact that it is a secreted protein that could be administered in vivo in a controlled manner depending on the clinical situation, render DEL-1 an attractive and promising prognostic and/or therapeutic target in neonatal sepsis.

## Methods

### Mice and monitoring

C57BL/6 mice of neonate (4–6 days old) or adult age (8–10 weeks old) were used as well as *Edil3*$^{-/-}$ mice (designated *Del1*$^{-/-}$) in C57BL/6 background[19]. Mice were kept in the animal house of the University of Crete, School of Medicine, in a temperature-controlled room and 12-hour light/dark cycle, with free access to standard laboratory chow and water. Non-breeding adult mice were maintained on standard rodent food (4RF21, Mucedola, Milano, Italy), and breeding adult mice were on a high-protein diet (4RF25, Mucedola, Milano, Italy). To achieve synchronized births of neonate mice, paired matings of WT and *Del1*$^{-/-}$ mice were placed weekly; pregnant females were isolated from males and followed closely thereafter three times per week until their date of birth. For all experimental procedures, neonates were removed from their mothers as a group and placed on a warming blanket. After the end of procedure, littermates remained with their mother till the end of the experiment. Both male and female mice were used for experiments. Sex was not tracked as a biological variable, as no difference related to DEL-1 expression or immune function were observed between the male and female neonate mice.

All animal experimentation was in adherence to the "NIH Guide for the Care and Use of Laboratory Animals" and all animal procedures were in accordance with institutional guidelines and were approved by the University of Crete's Animal Care and Use Committee and the Veterinary Department of the Heraklion Prefecture (license number 150760 and 6540).

### Murine polymicrobial sepsis

Polymicrobial sepsis was induced using the fecal peritonitis method[27,73,74]. Specifically, the cecum contents of adult C57BL/6 mice (6–8 weeks old) were suspended in 5% dextrose solution at a final concentration of 80 mg/ml and then passed through a 70-nm filter. The suspension was aliquoted and stored in 15% containing glycerol stocks at −80 °C for up to three months until challenge. To account for batch and/or donor variation, each CS preparation was first tested in control adult and neonate mice to reach the desired lethal dose. The same batch of CS and challenge dose was kept constant among different mouse genotypes and ages for an entire experiment. The amount of 1.1 to 1.6 mg/per gram of mouse bodyweight of CS (depending on the intended severity of sepsis) was injected intraperitoneally (i.p.) in WT or *Del1*$^{-/-}$ mice of neonatal or adult age. Age- and gender-matched mice received equal amount of 5% dextrose and served as controls.

### In vivo DEL-1 and anti – IL-10 receptor treatments

A group of *Del1*$^{-/-}$ newborn septic mice was treated with recombinant DEL-1 protein fused with human IgG Fc (DEL-1-Fc). The DEL-1-Fc protein was prepared as follows: The gene encoding the full-length DEL-1 protein sequence was amplified by polymerase chain reaction (PCR) and cloned into the HindIII and KpnI sites of the mammalian expression vector pSecTag2 (InVitrogen, Waltham, Massachusetts, USA), which contains an N-terminal secretory tag (murine Igk leader sequence) and a C-terminal polyhistidine tag (His-tag)[22,24]. The human IgG Fc gene was cloned between *Del1* and His-tag at EcoRI and XhoI sites and in frame with both sequences. DEL-1 was then expressed as a soluble Fc-fusion protein secreted into the culture medium of transfected HEK-293F suspension cells (InVitrogen, Waltham, Massachusetts, USA). DEL-1-Fc protein was then purified by Ni-affinity chromatography via loading of concentrated culture supernatants onto a His-Trap column (GE Healthcare, Chicago, USA) connected to an ÄKTA-FPLC system (GE Healthcare, Chicago, USA)[22,24]. The identity and purity of the protein were confirmed using immunoblotting and SDS-PAGE[22,24]. Five μg DEL-1-Fc per mouse was administered once intravenously (i.v.; orbital vein), 15 min prior to the induction of sepsis. Age-matched newborn mice received the same amount of

recombinant IgG1- Fc (Recombinant IgG1 Fc protein carrier free, catalog no. 110-HG, R&D Systems, Minneapolis, USA) and served as controls.

Another group of WT or $Del1^{-/-}$ newborn septic mice were treated with purified endotoxin-free anti-mouse IL-10R blocking antibody (Purified anti-mouse CD210 antibody, low endotoxin, azide-free, clone 1B1.3a, catalog no. 112710, Biolegend, San Diego, CA, USA). Five µg of IL-10R blocking antibody per mouse was administrated once i.v. (orbital vein), 15 min prior to the induction of sepsis. Mice were briefly anesthetized by sevoflurane inhalation prior to orbital vein injections. Age-matched newborn mice received the same amount of control IgG1 (Purified Rat IgG1, κ Isotype Ctrl Antibody, low endotoxin, azide-free, clone RTK2071, catalog no. 400432, Biolegend, San Diego, CA, USA) and served as controls. In a different set of experiments, the antibiotic meropenem at a concentration of 25 µg/g was administered i.p. 2 hours after CS-induced sepsis.

### In vivo murine sample collection, analysis, and survival experiments

At specific time points following CS administration, mice were anaesthetized with i.p. ketamine (100 mg/kg) and xylazine (8 mg/kg) and peritoneal lavage, whole blood, bone marrow and tissues were harvested for analysis. Peritoneal lavage was collected by i.p. injection of 600 µl of sterile PBS (Gibco, Invitrogen, Carlsbad, CA, USA) per newborn mouse and 1.5 ml per adult mouse, and then by gently massaging the body cavity 20 times before aseptic aspiration. Blood was drawn via cardiac puncture into a 25-gauge insulin syringe containing 20 µl heparin (1000 U/ml) and placed on ice. Prior to tissue collection, mice were perfused with ice-cold PBS via the right ventricle. To isolate bone marrow cells, femoral bones were removed and transferred into sterile petri dishes. Bone marrow was flushed out of femurs, using a 23-gauge needle. In another set of experiments, sepsis survival of WT and $Del1^{-/-}$ mice, and WT mice that received either IgG-Fc or IL-10R blocking antibody or DEL-1-Fc was evaluated.

### Study subjects

For all analyses on human samples, informed, written consent was obtained from all participants at the time of the recruitment, and the studies were conducted in accordance with the Helsinki Declaration ethical standards. No compensation was provided to participants included in this study. All procedures were conducted upon approval of the Institutional Review Board of the University General Hospital of Heraklion (approval numbers 1724, 2418, and 375,047).

The human serum used in this study was obtained from the following sources: First, human serum from the cord blood of healthy newborns and four-year-old healthy children were kindly provided by Prof L. Chatzi from "the Mother–Child" birth cohort of the University of Crete. Secondly, serum from adult septic patients (>18 years old), and neonate septic patients (0–28 days old), admitted to the neonatal and adult Intensive Care Units of the Heraklion University hospital were obtained. Serum from sex and age-matched healthy adult volunteers, sex and age-matched healthy newborns were used as controls. Both male and female subjects were enrolled. Sex was not tracked as a biological variable, as no difference related to DEL-1 expression, especially in neonatal age, was observed between male and female subjects.

Neonatal sepsis was defined as neonates (age <28 days on admission) with presence of at least one clinical sign (temperature instability, cardiovascular or respiratory instability, skin symptoms, gastrointestinal symptoms) and at least one laboratory result which is suggestive for neonatal sepsis (white blood cell count, platelet count, C- reactive protein, absolute neutrophil count) and elevated IL-6 > 50 pg/dl[57,58]. Adult sepsis was defined as adults (>18 years old) with presence of evidence of infection (possible or confirmed) and at least two out of four SIRS criteria (systemic inflammatory response syndrome) or two-point increase in SOFA (sequential organ failure assessment) score. Additional information such as age, sex, gestational age (for neonates), blood neutrophil counts, and outcome (length of stay and 28-day mortality) were collected.

### Measurement of myeloperoxidase (MPO) activity

For MPO determination, tissues were homogenized in 50 nmol/l phosphate buffer, pH 6.0, with 0.5% hexadecyltrimethylammonium bromide using mortar and pestle. Samples were frozen/thawed three times, centrifuged for 10 min at 10,000 $g$, and supernatants were stored at −20 °C. To determine MPO, in 96-well plates, 10 µl of sample were added to 190 µl assay buffer containing phosphate buffer 50 mM, pH 6.0 and 0.167 mg/ml o-dianisidine (Sigma-Aldrich, St Louis, USA) and 0.0005% $H_2O_2$[75]. Absorbance at 450 nm (A450 nm) was measured in a microplate reader at 15 min. Results were normalized per protein content of tissues and were reported as arbitrary units per mg of tissue.

### Determination of leukocyte infiltration and bacteria clearance in vivo

Total white blood cell (WBC) counts of cells isolated from peritoneal lavage, whole blood and bone marrow were estimated by hemocytometer, whereas differential counts were determined by flow cytometry (see below). Peritoneal and whole blood bacterial counts were determined by culturing 100 µl of serially diluted peritoneal lavage or whole blood sample on Luria–Bertani (LB) agar plates at 37 °C overnight. Plates were counted after 24 hours of plating. Bacterial counts in tissues were determined as follows: whole organs were briefly immersed in 70% ethanol and sterile water, and then disrupted with mortar and pestle. The homogenized tissues were suspended in 500 µl of sterile PBS. Suspensions were serially diluted, plated on LB agar plates and incubated at 37 °C for 24 h. Results are reported in colony-forming units (CFU) per milliliter of blood or CFU per tissue.

### Regulation of DEL-1 expression by cytokines in HUVECs and Ea.hy929 cells

HUVEC cells were purchased from Lonza (Basel, Switzerland) and Ea.hy926 endothelial cell line was kindly provided by Prof. Kardasis laboratory, University of Crete, School of Medicine. Cells were cultured in Endothelial Cell Growth Medium with Supplement Mix (PromoCell, Heidelberg, Germany). Cells were seeded in 12 well plates (300,000 per well) and on the next day they were treated with 20 ng/ml human IL-6, 20 ng/ml human IL-10, or 20 ng/ml human IL-17A (PeProtech, London, UK) for 4, 8, and 24 hours in basal Endothelial Cell Growth Medium without supplements. Total RNA was isolated and quantitative PCR was performed as described below.

### Mesenchymal stromal cell isolation and tissue culture

MSCs isolated from Wharton jelly, were cultured in fresh Dulbecco modified Eagle Medium, (DMEM) low glucose (1 g/L) supplemented with 10% fetal bovine serum (FBS) and 1% (100 IU/mL) penicillin-streptomycin at 37 °C. Culture medium was replaced twice per week; when MSCs reached 80–90% confluency, they were detached using 0.25% Trypsin–1 mM EDTA (Gibco, Invitrogen, Carlsbad, CA, USA). Cells were then reseeded at a concentration of 2000–3000 cells/cm$^2$ and further expanded for a total of 12 passages (P12). Isolation and culture of MSCs was performed after informed consent and has been approved by Ethics Committee of the University Hospital of Heraklion, Crete, Greece (approval number 1724).

### RNA isolation and quantitative PCR

Total RNA was extracted from adults and tissues, human MSCs, HUVEC and Ea.hy926 human endothelial cells using TRIzol reagent (Life Technologies, Carlsbad, CA, USA) and quantified by spectrometry at 260 and 280 nm. One microgram of total DNA-digested RNA was used

for cDNA synthesis (Thermoscript RT; Invitrogen, Carlsbad, CA, USA). The SYBR Green method was followed in the PCR reaction.

The murine and human primers that were used in the PCR reaction are the following: mouse DEL-1; forward (fwd): 5′-CCTGTGAGATA AGCGAAGC-3′ and reverse (rev): 5′-GAGCTCGGTGAGTAGATG-3, mouse IL-17A; fwd: 5′-CCGTTCCACGTCACCCTGGAC-3′ and rev: 5′-GG TCCAGCTTTCCCTCCGCATTG-3′, mouse IL-10; fwd: 5′- GCGCTGTCA TCGATTTCTCCCCTG-3′ and rev: 5′-GGCCTTGTAGACACCTTGGTC TTGG-3′, mouse ICAM-1; fwd: 5′-GGTTCTCTGCTCCTCCACAT-3′ and rev: 5′ -CCTTCCAGGCTTTCTCTTTG-3′, mouse Ribosomal Protein S9 (RSP9) (housekeeping); fwd: 5′-GCTAGACGAGAAGGATCCCC-3′ and rev: 5′-. CAGGCCCAGCTTAAAGACCT -3′, human DEL-1; fwd: 5′- TGAT TACCCAAGGAGCCAAG- 3′ and rev: 5′- TCGCAAAGTGCAATGTCTTC- 3′ and human Glyceraldehyde 3-phosphate dehydrogenase (GAPDH); fwd: 5′-CGACCACTTTGTCAAGCTCA- 3′ and rev: 5′-AGGGGTCTACA TGGCAACTG- 3′.

Annealing was carried out at 60 °C for 30 s, extension at 72 °C for 30 s, and denaturation at 95 °C for 30 s for 40 cycles using the ABI 7500 System, according to the manufacturer's protocol (Applied Biosystems, Waltham, Massachusetts, USA). Analysis of the fold change was performed based on the Pfaffl method[76].

## Murine and human serum cytokine, G-CSF, SDF-1, and DEL-1 measurement

Cytokine concentration of IL-6, TNF, IL-10, CXCL-1, and IL-17A in mouse and human serum was determined by enzyme-linked immunoabsorbent assay (ELISA) at the indicated time points using ELISA kits (ELISA Max Deluxe assays, Biolegend, San Diego, CA, USA), according to manufacturer's instructions. G-CSF and CXCL12/SDF-1 were measured in serum or bone marrow extracts respectively using mouse ELISA (G-CSF and Mouse CXCL12/SDF-1 alpha Quantikine ELISA, RnD Systems, Minnesota, USA), according to the manufacturer's instructions. Mouse tissue DEL-1 was measured using a validated commercially available assay (Mouse EDIL3 ELISA kit, CUSABIO, Houston, USA). Human serum DEL-1 was measured using a validated commercially available assay (Human EDIL3 DuoSet ELISA, RnD Systems, Minnesota, USA). Triton 0.5% was added to the serum samples prior to analysis.

## Neutrophil and GMP analysis by flow cytometry

Total white blood cell (WBC) counts of peritoneal lavage isolated cells, whole blood, and bone marrow cells were isolated from mice, assessed by hemocytometer counting using acetic acid 3% treatment, and placed in flow cytometry staining buffer. For neutrophil analysis, cells were first incubated with Anti-Mouse CD16/CD32 (Mouse Fc Block) (1:200, catalog no. 553141, BD Pharmigen, San Diego, CA, USA) and then staining was performed using PE anti-mouse Ly6G (1:200, 1A8-Ly6g, catalog no. 12-9668-82, E-bioscience, San Diego, CA, USA) or PE-anti- mouse Ly6G (1:200, 1A8, catalog no. 127608, Biolegend, San Diego, CA), and APC anti-mouse CD11b (1:100, M1/70, catalog no. 101229, Biolegend, San Diego, CA, USA). Further analysis of monocytes and myeloid-derived suppressor cells was performed with FITC anti-mouse CD11b antibody (1:100, M1/70, catalog no. 101205, Biolegend, San Diego, CA, USA), PE anti-mouse Ly6G (1:200, 1A8-Ly6g, catalog no. 12-9668-82, E-bioscience, San Diego, CA, USA) or PE- anti- mouse Ly6G (1:200, 1A8, catalog no. 127608, Biolegend, San Diego, CA), APC anti-mouse Ly6C (1:200, HK1.4, catalog no. 128016, Biolegend, San Diego, CA, USA) and PerCP/Cy5.5 anti-mouse CD11c (1:200, N418, catalog no. 117328, Biolegend, San Diego, CA, USA). Staining with PE anti-mouse CD51 (1:200, RMV-7, catalog no. 104105, Biolegend, San Diego, CA, USA) and FITC anti-mouse CD61 (1:100, catalog no. 10430, Biolegend, San Diego, CA, USA) were used for integrin expression analysis in the total bone marrow cells. The appropriate isotype controls were used in each case.

For GMP cell analysis, bone marrow total cells were collected as described above. Erythrocytes were removed via incubation with ACK

lysing buffer (Thermofisher, Invitrogen, Carlsbad, CA, USA). Cell suspension was then counted as described above and was kept frozen in FBS supplemented with 10% DMSO at −80 °C and then in liquid nitrogen for up to 1 month prior to use. For flow cytometry analysis, bone marrow cells were washed and were stained with PE-Cy7 anti-mouse Sca-1 antibody (1:100, D7, catalog no. 108113, Biolegend, San Diego, CA, USA), PE anti-mouse CD34 (1:50, MEC14.7, catalog no. 119307, Biolegend, San Diego, CA, USA), APC anti-mouse Lineage antibody cocktail (1:10, catalog no. 558074, BD Pharmigen, San Diego, CA, USA), BV480 anti-mouse CD117 (c-Kit) (1:100, 2B8, catalog no. 566074, BD Pharmigen San Diego, CA, USA) and BV421 anti-mouse CD16/32 (1:50, catalog no. 101331, Biolegend, San Diego, CA, USA) for 4 hours in 4 °C. The proper isotype controls were used in each case. GMP progenitor cells were identified as live cells, Lineage neg, CD117 (c-Kit) positive, Sca-1 negative, and CD16/32 and CD34 positive. For hematopoietic stem and progenitor cell (HSPC) analysis, a lineage antibody cocktail (Lin) (1:10, catalog no. 558074, BD Pharmigen, San Diego, CA, USA), BV480 anti-mouse CD117 (c-Kit) (1:100, 2B8, catalog no. 566074, BD Pharmigen San Diego, CA, USA), PE-Cy7 anti-mouse Sca-1 antibody (1:100, D7, catalog no. 108113, Biolegend, San Diego, CA, USA), Alexa Fluor 700 anti-mouse CD48 (1:50, HM48-1, catalog no. 103425, Biolegend, San Diego, CA), PerCP/Cy5.5 anti-mouse CD150 (SLAM) (1:50, TC15-12F12.2, catalog no. 115921, Biolegend, San Diego, CA) and PE- anti-mouse CD135 (Ftl3) (1:50, A2F10, catalog no. 135305, Biolegend, San Diego, CA) were used. Gating strategies for HSPC were as follows: LSK, Lin−Sca-1+cKit + ; LT-HSC, CD48 − CD150 + LSK; ST-HSC, CD48−CD150−LSK; MPP, CD48 + CD150−LSK; MPP2, Flt3 − CD48 + CD150 + LSK; MPP3, Flt3−CD48 + CD150−LSK[33,35]. The flow cytometry events were acquired in a FACS Canto II instrument (BD Biosciences, San Jose, CA) and analyzed with the use of FlowJo v10.7.1 Software (BD Biosciences, San Jose, CA, USA).

## Blood neutrophil phagocytosis assay

Ex vivo blood neutrophil phagocytosis assay was performed using the Phagotest kit (BD Biosciences, San Jose, CA, USA). Briefly, 50 µl of heparinized whole blood isolated from animals, was incubated soon after collection, with 10 µl of FITC-E.coli (non-opsonised) at 37 °C for 10 min, based on manufacturer instructions. Following incubation tubes were placed on ice to terminate phagocytosis. Red blood cells were lysed, cells were washed twice and incubated with APC anti-mouse CD11b (1:200, M1/70, catalog no. 101211, Biolegend, San Diego, CA, USA) and PE anti-mouse Ly6G (1:200, 1A8, catalog no. 127608, Biolegend, San Diego, CA, USA) for 15 min at 4 °C. Cells were washed and fixed in 2% paraformaldehyde. Cells were visualized in a FACS Canto II instruments (BD Biosciences, San Jose, CA, USA) and data were analyzed using FlowJo v10.7.1 Software (BD Biosciences, San Jose, CA, USA).

## Statistical analysis

All data measurements were taken from distinct biologically independent samples. All numeric data were evaluated for normality using the Kolmogorov–Smirnov test or Shapiro–Wilk test. The numerical data that passed the normality test and the PCR results were analyzed using unpaired two-sided Student's $t$ test or one-way ANOVA with the Bonferroni multiple-comparison post-test and were expressed as mean ± standard deviation (SD). Comparison of measurements that failed normality tests was performed using the Mann–Whitney $U$ test or the Kruskal–Wallis test with the Dunn multiple-comparison post-test and were expressed as median with interquantile range. Categorical data were analyzed by Fisher's exact or Chi-squared test. Kaplan-Meier curves were performed for survival experiments, and survival curves were compared between groups using a log-rank test. The GraphPad InStat software (GraphPad 8.0, San Diego, CA, USA) was used for analysis. $P$ values < 0.05 were considered significant. Results are representative of at least three independent experiments.

## Reporting summary

Further information on research design is available in the Nature Portfolio Reporting Summary linked to this article.

## Data availability

Source data are provided with this paper.

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

## Acknowledgements

This work has been funded by the Hellenic Foundation for Research and Innovation grant to Eleni Vergadi (HFRI, General Secretariat for Research and Technology, GSRT Grant No 1010) and by US National Institutes of Health grants (DE026152 and DE028561) to G.H.; T.C. is supported by the Deutsche Forschungsgemeinschaft (TRR-332, Project B4). O.K. was supported by the Greek State Scholarships Foundation and the European Union (European Social Fund) through the program "Human Resource Development, Education and Lifelong Learning", 2014-2020, in the context of the Act "Support of human resources via doctoral research 2: IKY Scholarship Program for Ph.D. Candidates of Greek Universities". We cordially thank Prof. Kardasis for his kind donation of Eah929 cell line and C. Vlata for providing expert assistance in flow cytometry data acquisition. We would also like to thank Prof L. Chatzi and Dr E Korakaki for their contribution to the provision of human samples. We extend our thanks to the staff at the animal care facility at the School of Medicine of the University of Crete and to all students and volunteers who have helped over the years, specifically to A. Batsali, E. Neofotistou-Nefeli, M. Danopoulou, N. Androulaki, S. Baiba, and M. Vretzou.

## Author contributions

E.V. conceived and designed the project, performed experiments, analyzed and interpreted data, and wrote the manuscript. O.K., K.L., E.I., I.L., V.I.A. performed experiments, analyzed and interpreted data; E.D., K.V., E.H., H.P. obtained human samples and interpreted data, E.G. supervised research, and revised the manuscript, G.H. and T.C. provided critical resources, designed experiments, interpreted data and revised the manuscript, and C.T. supervised research, designed experiments, interpreted data and revised the manuscript. All authors contributed to the writing of the manuscript.

## Competing interests

The authors declare no competing interests.
