## [Peer Review File · Nature Communications]

REVIEWER COMMENTS

Reviewer #1 (Remarks to the Author):

NCOMMS-21-42218

This a novel and potentially important study performed by Dr. Vergadi et al., whose goal is to evaluate the role of the developmental endothelial locus-1 (DEL-1), which is a known endothelial-derived anti-inflammatory factor, in the neonatal immune response to sepsis. As the authors state, newborns are a vulnerable population prone to infections due to (mainly) an immune system fundamentally different from children and adults.

The authors propose that a dysregulated expression and function of DEL-1 may be responsible for the dysfunctional recruitment of leukocytes to sites of inflammation in the newborns, and also evaluated the prognostic value of DEL-1 in the management of neonatal sepsis. The authors demonstrate that DEL-1-deficient neonatal mice are more vulnerable to mortality to polymicrobial sepsis due to a reduced number of neutrophil progenitors in the bone marrow, which leads to neutropenia and increased bacteremia. Furthermore, they found that DEL-1 over-expression in the neonatal period is attributed to a high IL-10/IL-17 ratio. While this manuscript does offer valuable mechanistic insights into the neonatal immune response to sepsis, there are some concerns outlined below that need to be addressed.

In general, this is a rather superficial and incomplete analysis of both the role of DEL-1 in neonatal sepsis and its regulation by the IL-10/IL-17 axis. As documented below, there is little insights into the mechanisms by which DEL-1 confers protection in sepsis. There is a great deal of descriptive data regarding the expression and concentrations of DEL-1, but less regarding how DEL-1 protects murine and human sepsis subjects. The key question is how does DEL-1 regulate bone marrow myeloid cell accumulation and release. Does it act similar to SDF-1 (CXCL12); does it act through CSF and progenitors, or is the target MSCs. These mechanistic questions need consideration.

The testing of IL-6, IL-10 and IL-17 on HUVECs seems to be going in a different direction and not focusing on the bone marrow, where the release of granulocytes and emergency myelopoiesis is regulated. All of the interactions between IL-10 and IL-17, and DEL-1 are by association (with the exception of the Fc studies discussed below). This reviewer is a little surprised that IL-10 or IL-17 KO were not used. The use of a neutralizing IL-10 Fc immunoadhesin is a good start, but should be used to further explore the interaction in the DEL-1 null mice. The combination of DEL-1 null mice and antiIL-10 Fc could have been used to identify the signaling pathways in the bone marrow responsible for the granulocyte response.

Also, as discussed below, the reliance on GR1+CD11b to phenotype the granulocytic population is a weakness. Gabrilovich and others have shown that neonates have expansion of MDSC subsets, and differentiation between MDSCs and mature PMNs and monocytes/macrophages seems essential.

Comments

1. Human results should be presented in different figures and not mixed with the results found in the animal models. More data regarding the human studies should be presented in the main manuscript body and not as supplemental figures.
2. It is known that DEL-1 is deposited in the endothelium and extracellular matrix and that degradation of the endothelial glycocalyx occurs during sepsis. In a murine model of sepsis, shedding of the endothelial glycocalyx and increased circulating DEL-1 expression has been documented, as well as correlated with sepsis severity. Circulating DEL-1 determination should be included in the study in addition to the tissue determinations reported.
3. What is the objective to compare the expression of DEL-1 in the different tissues evaluated in the neonatal mice (Figure 1A)? Figures 1B and 1C show more relevant information (neonatal vs. adult mice). It is clear that DEL-1 is over-expressed in intestine, lung and kidney; nevertheless, why the authors presented one time point for intestine and brain (adult vs. 4 days old pups) while for lung and kidney they showed DEL-1 expression over-time (Figure 1C), not including 4 days old pups?
4. Did the authors determine the expression of DEL-1 in the spleen?
5. DEL-1 mRNA expression is valuable; nevertheless, tissue and circulating DEL-1 (protein) expression should be also determined (see comment #2).
6. Regarding the human data presented in Figure 1D, the authors stated that they found a significant increase of DEL-1 in neonates compared with children. Why the authors mixed preterm and full-term neonates (34w-40w)? Why did they compare newborns with children and not with adults? Is there any role of pregnancy on DEL-1 expression as the authors use cord blood samples? This should be addressed in the manuscript.
7. Figure 2A shows an increased DEL-1 mRNA expression in lung in neonatal mice which seems to correlate with increased bacteremia; nevertheless, in kidney and intestine DEL-1 mRNA expression did

not change and they still found a significant increase of bacteria CFU. Similarly, there is no difference in MPO activity in the neonatal mice after sepsis, which suggests the presence of dysfunctional granulocytes. Were the neutrophils quantified in the different compartments evaluated (this is critical as DEL-1 has a key role in leukocyte infiltration)? How the authors interpret the findings in the peritoneal lavage?

8. It is unclear why the author made different determinations in different tissues. This reviewer would expect to see DEL-1 mRNA expression, bacterial CFU and neutrophil quantification in lung (increased DEL-1 after sepsis in neonatal mice) or any other tissue that the authors consider relevant for this study. This same comment applies for Figure 3.

9. It is clear (Fig. 5) that *Del1*^{-/-} null mice have BM exhaustion after sepsis that is reversed after administration of DEL-1-Fc in blood. Did the authors quantify the infiltration of neutrophils in and bacteria CFU in the peritoneal cavity after DEL-1-Fc administration? With this data would be possible determine the role of DEL-1 in the infiltration of leukocytes to the site of infection and subsequent phagocytosis activity (which is known to be deficient in this model).

10. The authors phenotype neutrophils as Gr1⁺Cd11b⁺ cells. While the anti-Gr1 antibody detects the Ly6G antigen (granulocytes), this antibody may cross-react with the antigen Ly6C (expressed widely in monocytes/macrophages).

In conclusion, it is still unclear if DEL-1 role in neonatal sepsis is positive as induces granulopoiesis, or negative as inhibits infiltration of leukocytes and the subsequent bacterial clearance?

Minor comments

1. The introduction lacks of background about "IL-10/DEL-1 axis". Role of IL-10 in neonatal sepsis.

2. The authors evaluated DEL-1 mRNA relative expression in multiple organs and they found that DEL-1 is over-expressed in neonatal mice (compared to adult mice) in lung, kidney and intestine. Although neonatal DEL-1 expression in the brain was significantly higher than the other organs evaluated, its expression was significantly low compared to adult DEL-1 expression in the brain. This is an interesting finding that could be addressed in the discussion.

Reviewer #2 (Remarks to the Author):

The manuscript by Vergadi et al entitled "A novel IL-10 – DEL-1 axis promotes granulopoiesis and sepsis survival in early life" revealed a novel role for Del-1 in promoting granulopoiesis and such effect may augment neonatal protection against sepsis in mice and human. Overall, the findings are important and with clinical significance. Some suggestions are listed for improved clarity in terms of underlying mechanisms.

- 1). Fig 1, 2. It would be more informative if the authors could identify the cellular source of elevated Del1 in the sepsis model (e.g. endothelial cells?).
- 2). Fig 2. For the measurement of MPO reflecting tissue damage due to neutrophil infiltration, it would be informative to have a later time point (e.g. 24 h) to correlate with sepsis severity. Likewise, the levels of tissue neutrophils 24 h post CS injection would be informative as well (fig 3).
- 3). Fig 6. The authors showed Del1 level alterations in a short time course (0, 6, 12 h). However, the authors only showed data for the 12 h time points regarding GMP levels. A time course depicting the progression of CMP, GMP levels in the bone marrow niche should be provided.
- 4). To better justify the focused study on neutrophils and granulopoiesis, the authors should rule out (or in) the effects of Del1 on monocytes/macrophages. Did Del1 impact monocyte/macrophage levels during neonatal sepsis?
- 5). Ex vivo phagocytosis assays of neutrophils impacted by Del1 would further strengthen the conclusion of this report.

Response to Reviewers

Manuscript Number: NCOMMS-21-42218

“A novel IL-10 – DEL-1 axis promotes granulopoiesis and sepsis survival in early life”

We would like to thank the reviewers for the constructive comments that helped us improve our manuscript. According to the reviewers’ suggestions, we have performed additional experiments to address their concerns. A point-by-point response to the reviewers’ comments is listed below.

Reviewer 1:

Comment: *“This a novel and potentially important study performed by Dr. Vergadi et al., whose goal is to evaluate the role of the developmental endothelial locus-1 (DEL-1), which is a known endothelial-derived anti-inflammatory factor, in the neonatal immune response to sepsis. As the authors state, newborns are a vulnerable population prone to infections due to (mainly) an immune system fundamentally different from children and adults. The authors propose that a dysregulated expression and function of DEL-1 may be responsible for the dysfunctional recruitment of leukocytes to sites of inflammation in the newborns, and also evaluated the prognostic value of DEL-1 in the management of neonatal sepsis. The authors demonstrate that DEL-1-deficient neonatal mice are more vulnerable to mortality to polymicrobial sepsis due to a reduced number of neutrophil progenitors in the bone marrow, which leads to neutropenia and increased bacteremia. Furthermore, they found that DEL-1 over-expression in the neonatal period is attributed to a high IL-10/IL-17 ratio. While this manuscript does offer valuable mechanistic insights into the neonatal immune response to sepsis, there are some concerns outlined below that need to be addressed.*”

In general, this is a rather superficial and incomplete analysis of both the role of DEL-1 in neonatal sepsis and its regulation by the IL-10/IL-17 axis. As documented below, there is little insights into the mechanisms by which DEL-1 confers protection in sepsis. There is a great deal of descriptive data regarding the expression and concentrations of DEL-1, but less regarding how DEL-1 protects murine and human sepsis subjects. The key question is how does DEL-1 regulate bone marrow myeloid cell accumulation and release. Does it act similar to SDF-1 (CXCL12); does it act through CSF and progenitors, or is the target MSCs. These mechanistic questions need consideration.”

Response: We thank the reviewer for the valuable comments. We agree with the reviewer that some mechanistic aspects were not investigated in the original submission of this article, however these aspects were addressed in previous studies. Indeed, the mechanisms whereby DEL-1 regulates bone marrow myeloid cell accumulation and release have been described in a previous study by some of the co-authors (Mitroulis et al; PMID: 28846069) and have also been reviewed (Hajishengallis et al; PMID: 30885428). Briefly, DEL-1 is expressed by several major cellular components of the hematopoietic niche, and has been shown to drive myelopoiesis in the bone marrow, by regulating HSC proliferation and myeloid lineage commitment of myeloid progenitors via $\beta 3$ integrin (Mitroulis et al, PMID: 28846069). Moreover, DEL-1 was found to contribute to myelopoiesis both at steady state as well as to emergency myelopoiesis upon G-CSF administration or LPS-driven inflammation (PMID: 28846069). These

basic mechanistic aspects underlie the novel and medically important direction of the present study, which identified a hitherto unidentified IL-10–DEL-1 axis that supports emergency granulopoiesis and thereby prevents neutropenia to enhance sepsis survival in early life.

To provide to the readers information on the mechanisms by which DEL-1 regulates myelopoiesis, we added this information in the introduction and discussion sections of the revised manuscript (lines 78-81, page4 and lines 377-381, page17 of the revised manuscript).

Comment: *The testing of IL-6, IL-10 and IL-17 on HUVECs seems to be going in a different direction and not focusing on the bone marrow, where the release of granulocytes and emergency myelopoiesis is regulated. All of the interactions between IL-10 and IL-17, and DEL-1 are by association (with the exception of the Fc studies discussed below). This reviewer is a little surprised that IL-10 or IL-17 KO were not used. The use of a neutralizing IL-10 Fc immunoadhesin is a good start, but should be used to further explore the interaction in the DEL-1 null mice. The combination of DEL-1 null mice and antiIL-10 Fc could have been used to identify the signaling pathways in the bone marrow responsible for the granulocyte response.*

Response: The reviewer makes a rational point here and has offered suggestions that helped us strengthen our conclusions. Since a major source of DEL-1 expression in the bone marrow niche is the endothelium (Mitroulis et al. JCI 2017 and Chen et al. Thrombosis Haemostasis 2018), we used HUVEC and Eah929 endothelial cells as an *in vitro* model, to first evaluate whether the neonatal cytokines have an impact on DEL-1 expression. From this experiment, we identified IL-17 and IL-10 as regulators of DEL-1 expression. The regulation of DEL-1 expression by IL-17 has been previously described by some of the co-authors (PMID: 26374165). As IL-10 was the dominant cytokine that was found to upregulate DEL-1 and was differentially regulated in neonates, we sought to further evaluate the role of IL-10, which has not been previously addressed in the context of DEL-1 regulation. IL-10 null mice have several defects, including developmental ones, that could confound the results, whereas the interventional approach with anti-IL-10, as suggested by the reviewer, only blocks the soluble cytokine and is thus appropriate for the purpose of this study. For this reason, we decided to address our research question with the use of a neutralizing IL-10 antibody. Specifically, by using anti-IL-10 neutralizing antibodies in the DEL-1 null mice, we evaluated the granulocyte response in the bone marrow. We found that IL-10 blockage in DEL-1 null mice did not reduce the granulocyte population in the bone marrow (Fig. 7h, of the revised manuscript), whereas the same treatment caused a reduction of the neutrophil pool in the bone marrow of WT neonates. This result clearly supports our hypothesis that IL-10 exerts its effect on bone marrow myelopoiesis via DEL-1, providing additional mechanistic evidence for DEL-1 function.

Comment: *Also, as discussed below, the reliance on GRI+CD11b to phenotype the granulocytic population is a weakness. Gabrilovich and others have shown that neonates have expansion of MDSC subsets, and differentiation between MDSCs and mature PMNs and monocytes/macrophages seems essential.*

Response: We thank the reviewer for this important comment. To address the reviewer's point, we also assessed the myeloid-derived suppressor cell (MDSC) subsets, to distinguish them from other granulocytes, in the bone marrow from healthy and septic WT and DEL-1 null mice. No effect of DEL-1 was noted on monocyte

populations or in MDSCs in neonates. We present these results in supplementary Fig. 2 and 7 of the revised manuscript.

Comment: *In conclusion, it is still unclear if DEL-1 role in neonatal sepsis is positive as induces granulopoiesis, or negative as inhibits infiltration of leukocytes and the subsequent bacterial clearance?*

Response: Similar to countless other examples of immune regulators with context-dependent effects, it is challenging to interpret if DEL-1's role in neonatal sepsis is positive as induces granulopoiesis, or negative as inhibits infiltration of leukocytes. However, our data have shown that DEL-1 restrains, rather than completely blocks, infiltration of leukocytes, suggesting a regulatory effect that may not necessarily be negative in the context of infection. Indeed, based on our findings, tissue leukocyte infiltration in neonates was negatively associated with sepsis survival, as DEL-1 null mice had increased neutrophilic infiltration in tissues but worse survival and, consistently, DEL-1 administration suppressed tissue neutrophil infiltration but improved survival from sepsis. In accordance with our observation, in a previous study in adult murine model of polymicrobial sepsis, ICAM-1-deficient mice demonstrated a significant reduction of mortality compared to WT mice. This is mentioned in the discussion, where we discuss that although leukocyte tissue recruitment is essential for pathogen elimination, it may provoke an overwhelming inflammatory response that may lead to microvasculature injury and contribute to organ damage and failure. To make it clearer for the readers, we further discuss in the revised manuscript that our data show that DEL-1 restrains the exaggerated neutrophil infiltration in DEL-1 null mice (lines 148-350, page 15 of the revised manuscript). Despite the fact that DEL-1 reduces infiltration of leukocytes into the site of infection, we demonstrate in this manuscript that its role in myelopoiesis is vital to support neutrophil production, while in the blood and sites of infection DEL-1 controls bacteremia and bacterial load.

Comment: 1. *Human results should be presented in different figures and not mixed with the results found in the animal models. More data regarding the human studies should be presented in the main manuscript body and not as supplemental figures.*

Response: In response to the reviewer's suggestion, we moved the human sepsis mortality data from the tables to main manuscript (Figure 5 in the revised manuscript), and we moved human results from Fig. 1, Fig. 2 and Fig. 5 to another figure (new Fig. 2 and new Fig. 5 in the revised manuscript). The human data of Fig. 7 were left in place as they support the mechanism that we would like to explain in this figure. In the revised manuscript, we added the word "human" in all subfigures of Fig. 7 that present human data, both in the figure and in the legend, to make sure that the reader will be able to distinguish them.

Comment: 2. *It is known that DEL-1 is deposited in the endothelium and extracellular matrix and that degradation of the endothelial glycocalyx occurs during sepsis. In a murine model of sepsis, shedding of the endothelial glycocalyx and increased circulating DEL-1 expression has been documented, as well as correlated with sepsis severity. Circulating DEL-1 determination should be included in the study in addition to the tissue determinations reported.*

Response: This is indeed an important point. To address this comment, we purchased a commercially available mouse DEL-1 ELISA kit and we tested serum of WT and neonatal mice upon steady state and sepsis. However, the commercially available murine DEL-1 ELISA has a detection limit of 28 pg/mL, which is higher than that of the human available assay (15.6 pg/mL). Circulating DEL-1 was apparently below the limit of detection of this assay in all the samples that were used. A specific signal for

DEL-1 protein was obtained only from tissue lysates. We are therefore unable to provide information on circulating DEL-1 in our study. However, besides the shedding of DEL-1 from the endothelial glycocalyx, as reported in the past, the main experimental evidence from the literature suggests that the pivotal role of DEL-1 occurs locally at the tissue level. Indeed, locally produced DEL-1 is mostly known for its functions in modulating tissue inflammation in the periphery via different mechanisms and myelopoiesis in the bone marrow. DEL-1 acts mostly locally and is a regulator of local tissue inflammation (PMID: 30885428, PMID: 34217213). Its functions mostly depend on the location of its tissue expression (PMID: 30885428). To better clarify this point, we have now specified that ‘tissue’ DEL-1 is elevated in neonates in Fig. 1 and Fig. 3 legend and in the text. Indeed, we have assessed DEL-1 levels in tissues lysates in the periphery (lung tissue) and in bone marrow (please see response to comment below).

Comment: 3. *What is the objective to compare the expression of DEL-1 in the different tissues evaluated in the neonatal mice (Figure 1A)? Figures 1B and 1C show more relevant information (neonatal vs. adult mice). It is clear that DEL-1 is over-expressed in intestine, lung and kidney; nevertheless, why the authors presented one time point for intestine and brain (adult vs. 4 days old pups) while for lung and kidney they showed DEL-1 expression over-time (Figure 1C), not including 4 days old pups?*

Response: We thank the reviewer for this comment. Indeed, the reason we compared DEL-1 expression in different tissues was to identify the tissues with higher DEL-1 expression, so that we could focus on studying DEL-1 regulation in tissue with abundant DEL-1 expression. The reason why we have experiments in neonates of 4 days and then time-course experiments in neonates is that these were different experiments addressing different questions. We initially performed experiments in all tissues with DEL-1 expression, in mice 4 days old vs adults. We show these 4 day-old mouse data in Fig. 1B in the revised manuscript. In this panel, we did not include spleen and liver as DEL-1 expression was very low in these tissues. Then, to understand DEL-1 expression kinetics in neonatal age, we addressed DEL-1 expression over time only in tissues that had substantial / high expression (such as lung and kidney). To be consistent, we performed again a new experiment with kinetics of DEL-1 in intestine over time and we included this in the revised manuscript. We have modified our figures (Fig. 1 and supplemental Fig. 1) to better present our findings based on the reviewer’s suggestions.

Comment: 4. *Did the authors determine the expression of DEL-1 in the spleen?*

Response: DEL-1 displays a selective tissue expression pattern. In response to the reviewer’s comment, we have now assessed the expression of DEL-1 in spleen. The expression in the spleen was very low compared to other tissues examined and this is depicted in the Fig. 1a in the revised manuscript. As the expression in the spleen was very low, we did not further exploit spleen DEL-1 levels for our subsequent experiments.

Comment: 5. *DEL-1 mRNA expression is valuable; nevertheless, tissue and circulating DEL-1 (protein) expression should be also determined (see comment #2).*

Response: The reviewer makes a valid point. As discussed above, we were unable to detect circulating DEL-1 in the serum of healthy and septic mice. However, we measured DEL-1 levels in lung and bone marrow tissue lysates from adult and neonate mice, under normal and septic conditions. The results are in accordance with mRNA findings and are depicted in Fig. 3 and 6 of the revised manuscript.

Comment: 6. *Regarding the human data presented in Figure 1D, the authors stated that they found a significant increase of DEL-1 in neonates compared with children. Why the authors mixed preterm and full-term neonates (34w-40w)? Why did they compare newborns with children and not with adults? Is there any role of pregnancy on DEL-1 expression as the authors use cord blood samples? This should be addressed in the manuscript.*

Response: Thank you for your comments. First, we included preterm and term neonates intentionally, to investigate whether gestational age had any impact on DEL-1 levels. No significant difference was found among preterm and term infants and this information is now mentioned in the revised manuscript. Secondly, we used two different sources of samples for our human data (this was mentioned in methods, but we made it more clear in the revised manuscript). We used human serum from the cord blood of 20 healthy newborns and 20 four-year-old healthy children that were obtained by the Mother - Child birth cohort available in our institution (RHEA cohort). The design of this birth cohort involved newborns (34w-40w) and children 4 years old. We opted to use samples from this birth cohort to verify that DEL-1 is elevated in human healthy newborns compared to older children. Then, to compare the DEL-1 kinetics in healthy and septic subjects of neonatal and adult age, we enrolled patients from the neonatal department, neonatal intensive care unit and adult intensive care unit together with (age matched?) healthy volunteer subjects. These neonatal samples that were obtained were not from cord blood. There is no information on the role of pregnancy on DEL-1 expression in cord samples. However, by including samples from two different sources, cord blood and also postnatal neonatal serum samples, we confirmed our results and minimized any potential impact that pregnancy might have had on DEL-1 cord levels. These issues are now clarified in Methods and Results sections of the revised manuscript.

Comment: 7. *Figure 2A shows an increased DEL-1 mRNA expression in lung in neonatal mice which seems to correlate with increased bacteremia; nevertheless, in kidney and intestine DEL-1 mRNA expression did not change and they still found a significant increase of bacteria CFU. Similarly, there is no difference in MPO activity in the neonatal mice after sepsis, which suggests the presence of dysfunctional granulocytes. Were the neutrophils quantified in the different compartments evaluated (this is critical as DEL-1 has a key role in leukocyte infiltration)? How the authors interpret the findings in the peritoneal lavage?*

Response: We thank the reviewer for this comment. It appears that the way the results were presented in Fig. 2 was confusing. Therefore, we tried to clarify these points in the revised manuscript.

In this part of the study, we aimed to show that DEL-1 is not suppressed upon sepsis in neonates, as it does in adults. We also aimed to correlate tissue DEL-1 levels with tissue neutrophilic response, prove that DEL-1 regulates tissue neutrophilic response and then correlate DEL-1 levels with tissue bacteria load, the latter as the result of neutrophilic infiltration.

First, we used MPO to investigate neutrophilic infiltration in tissues. Our main aim was to study neutrophil kinetics. Investigating neutrophil function and whether there is a difference among neonates and adults was beyond the scope of this study. Based on the literature, neutrophils may be dysfunctional in neonates, but MPO is equally functional in term neonates and adults, as it has been shown by *Levy O, et al*: “neutrophils from healthy term newborns and adults contain equal concentrations of the azurophilic granule proteins myeloperoxidase and defensin” (Levy O. PMID: 12032258). Thus, we believe that MPO is a reliable marker of neutrophilic infiltration in adult and neonatal mice in this model. We now mention this in the revised manuscript.

As far as the use of peritoneal lavage is concerned, we used neutrophil counts and bacterial counts in the peritoneal lavage, instead of intestine MPO and intestine bacteria load, as a more accurate indication of intestinal neutrophilic infiltration and inflammation. The peritoneum was also the site of infection in our model, thus we believe that studying neutrophilic infiltration and bacterial clearance in the peritoneum is important to assess host responses in this model. We clarified this in the revised manuscript.

Finally, we did not aim to directly correlate DEL-1 levels with tissue bacteria load. We show that in all tissues investigated (e.g., kidney and intestine), DEL-1 levels were not suppressed in neonates upon sepsis as they did in adult mice. Subsequently, MPO was increased in the lung and in the peritoneal lavage in adults, but not in neonates (lung) or was increased to a lesser extent in peritoneal lavage. The reviewer is right that this phenomenon was not observed in kidney, as we could not detect significant neutrophil infiltration even in adult mice. We also tested kidney samples collected at 12 hours of sepsis and still did not detect an increase in neutrophilic infiltration in the kidney. We believe that the degree of sepsis that we induced was not severe enough to provoke significant inflammation in distal organs such as kidney and for this reason we removed the kidney data from the revised manuscript.

To go beyond correlations, we went further to prove that DEL-1 controls tissue neutrophil infiltration in neonatal mice, by measuring MPO in the lung and neutrophil counts in the peritoneal lavage (Fig. 3 in the original manuscript). To make our statements clearer, we merged these figures in the revised manuscript (Fig. 3 of the revised manuscript). We also removed all bacterial load experiments from this figure (which are now shown in Fig. 5) to focus on our statement that DEL-1 is not suppressed in sepsis in neonates and controlled neonatal tissue neutrophil infiltration. Bacterial loads are now depicted in Fig. 4 of the revised manuscript. We thank again for this valuable comment that helped us improve the presentation of our results.

Comment: 8. *It is unclear why the author made different determinations in different tissues. This reviewer would expect to see DEL-1 mRNA expression, bacterial CFU and neutrophil quantification in lung (increased DEL-1 after sepsis in neonatal mice) or any other tissue that the authors consider relevant for this study. This same comment applies for Figure 3.*

Response: As explained above, we agree with the reviewer at this point. Indeed, determinations in tissues such as kidney, were irrelevant and confusing. For this reason, we streamlined the presentation of the results by removing kidney data from the main manuscript, and focused on lung and intestine (as peripheral tissues) and bone marrow (since this is also relevant in this study). We have also merged Fig. 2 and 3 of the original manuscript, in the new Fig. 3 in the revised manuscript.

Comment: 9. *It is clear (Fig. 5) that Del1^{-/-} null mice have BM exhaustion after sepsis that is reversed after administration of DEL-1-Fc in blood. Did the authors quantify the infiltration of neutrophils in and bacteria CFU in the peritoneal cavity after DEL-1-Fc administration? With this data would be possible determine the role of DEL-1 in the infiltration of leukocytes to the site of infection and subsequent phagocytosis activity (which is known to be deficient in this model).*

Response: We agree with the reviewer. Given that DEL-1-Fc abrogates the infiltration of leukocytes to the infection site, it is important to determine the outcome of bacterial load at the site of infection as well. We quantified the infiltration of neutrophils in the peritoneal cavity in Del1^{-/-} mice after DEL-1-Fc administration and this was depicted in Fig. 3 of the original manuscript (now Fig 3f of the revised manuscript). In this figure, we show that DEL-1-Fc administration reduces infiltration at the site of infection and

in peripheral tissues in DEL-1 null mice. However, even after DEL-1-Fc treatment, there is still substantial infiltration in the peritoneum of DEL-1 null mice at levels similar to that of WT mice, suggesting that DEL-1-Fc acts as a homeostatic molecule that abrogates the exaggerated neutrophilic infiltration in DEL-1 null mice. We then quantified the bacterial load in the peritoneal cavity at 12 hours post sepsis induction in DEL-1^{-/-} mice that received DEL-1-Fc, as the reviewer suggested, and we found that bacterial load was more effectively controlled in the mice that received DEL-1-Fc, despite the lower levels of leukocytes. This finding is in accordance with our results that DEL-1 not only controls bacteremia but can help in achieving control of bacteria load at the site of infection (source control) as well. Additionally, it appears that an exaggerated leukocyte infiltration is not necessary to achieve source control.

Finally, we tested the phagocytosis activity in our model. Term neonates are known to have comparable phagocytosis activity to this of adults. It has been shown though that DEL-1 attenuates complement-dependent phagocytosis through inhibition of complement receptor 3, aka Mac-1-integrin (PMID: 24352615). However, neonates are partially deficient in opsonins, such as complement and IgG, and in early life phagocytosis is performed via different non-opsonic receptors. To address whether DEL-1 affects phagocytosis in neonatal age, we performed ex vivo phagocytosis assay in neutrophils from whole blood. We infected neutrophils ex vivo, in the presence or absence of DEL-1-Fc, with unopsonised *E. coli* to recapitulate the phagocytosis conditions that occur in early life. We did not notice any difference in phagocytosis between adult WT or neonate WT mice treated with DEL-1-Fc compared with ones treated with IgG-Fc. These results are now presented in supplemental Fig. 4 in the revised manuscript.

Comment: 10. *The authors phenotype neutrophils as Gr1+ Cd11b+ cells. While the anti-Gr1 antibody detects the Ly6G antigen (granulocytes), this antibody may cross-react with the antigen Ly6C (expressed widely in monocytes/macrophages).*

Response: The reviewer makes a correct point here. We have used an anti-mouse Ly6G (Gr-1) antibody of clone 1A8 that detects only Ly6G (E Biosciences, Cat # 12-9668-82) (and not that of RB6-8C5 that may cross-react with Ly6C), minimizing any cross-reactivity with Ly6C. This was not specified in our manuscript and we thank the reviewer for noticing.

To address the reviewer's point and to test whether DEL-1 may have an impact in monocytes, we repeated experiments using CD11b, Ly6G and Ly6C in bone marrow, and peritoneal lavage of WT and DEL-1 null mice. As previously noted, no effect of DEL-1 was noted on CD11b⁺ Ly6C⁺ monocyte populations in neonates. We present these results in supplementary figure 2 of the revised manuscript.

Minor comments

Comment: 1. The introduction lacks of background about "IL-10/DEL-1 axis". Role of IL-10 in neonatal sepsis.

Response: We thank the reviewer for this suggestion. In the revised manuscript, we added background information in the introduction that there are no data on the regulation of DEL-1 by IL-10. Also, we added background information on the role of IL-10 in neonatal sepsis in the discussion section of the revised manuscript (page 4, lines 84-85 in the revised manuscript).

Comment: 2. *The authors evaluated DEL-1 mRNA relative expression in multiple organs and they found that DEL-1 is over-expressed in neonatal mice (compared to adult mice) in lung, kidney and intestine. Although neonatal DEL-1 expression in the brain was significantly higher than the other organs evaluated, its expression was*

significantly low compared to adult DEL-1 expression in the brain. This is an interesting finding that could be addressed in the discussion.

Response: We thank the reviewer for this suggestion. In the revised manuscript, we have discussed the finding regarding DEL-1 expression levels in brain tissue (lines 314-318, page 14 of the revised manuscript).

Reviewer 2:

The manuscript by Vergadi et al entitled “A novel IL-10 – DEL-1 axis promotes granulopoiesis and sepsis survival in early life” revealed a novel role for Del-1 in promoting granulopoiesis and such effect may augment neonatal protection against sepsis in mice and human. Overall, the findings are important and with clinical significance. Some suggestions are listed for improved clarity in terms of underlying mechanisms.

Comment: 1. *Fig 1, 2. It would be more informative if the authors could identify the cellular source of elevated Dell in the sepsis model (e.g. endothelial cells?).*

Response: In the present work, we investigated mostly lung and bone marrow, where the cellular source of DEL-1 was earlier reported to be broad and includes not only endothelial cells, but other stromal cells and some immune cells (macrophages). Specifically in the bone marrow, where the critical IL-10 – DEL-1 axis operates, the cellular source of DEL-1 has been shown to be those niche cells that have a major role in the maintenance of HSCs (i.e, arteriolar endothelial cells, osteoblastic lineage cells and predominantly perivascular mesenchymal stromal cells) (Mitroulis et al., JCI 2017). It is likely that all these niche cells contribute DEL-1 for stimulating emergency granulopoiesis in our model. Additionally, it is known that the mesenchymal stromal cell pool is expanded in early life (PMID 30700727), thus the increased numbers of mesenchymal stromal cells in the bone marrow niche of neonates may be the main source of high DEL-1 in the neonatal bone marrow. Therefore, we believe that the investigation of the cellular source of DEL-1 will not add significant information over what is already published in the literature. We have added the above information in the discussion to highlight the existing knowledge on the cellular sources of DEL-1.

Comment: 2. *Fig 2. For the measurement of MPO reflecting tissue damage due to neutrophil infiltration, it would be informative to have a later time point (e.g. 24 h) to correlate with sepsis severity. Likewise, the levels of tissue neutrophils 24 h post CS injection would be informative as well (fig 3).*

Response: We thank the reviewer for this suggestion. To reflect neutrophil kinetics over time among adult and neonates, we repeated the experiments at a later time point, at 12-hours following sepsis induction (later point than that of 6 hours that was initially used), and we included these data in the revised manuscript (new Fig. 3 in the revised version). Neutrophil kinetics over time for WT and Dell^{-/-} neonates are now depicted in Fig 5.

Comment: 3. *Fig 6. The authors showed Dell level alterations in a short time course (0, 6, 12 h). However, the authors only showed data for the 12 h time points regarding GMP levels. A time course depicting the progression of CMP, GMP levels in the bone marrow niche should be provided.*

Response: We thank the reviewer for this comment. We performed further experiments to analyze cell populations in the bone marrow at 24hours sepsis, and we have added the GMP quantification in Fig. 6 of the revised manuscript. We didn't go further than 24 hours, as we have a significant loss of animals due to sepsis after this point. We found no difference in the progression of CMP levels between WT and DEL-1 null mice.

Comment: 4. *To better justify the focused study on neutrophils and granulopoiesis, the authors should rule out (or in) the effects of Dell on monocytes/macrophages. Did Dell impact monocyte/macrophage levels during neonatal sepsis?*

Response: We thank the reviewer for this important comment. To address the reviewer's point, we performed additional flow cytometry analysis for CD11b+Ly6C+ monocytes in the bone marrow, and peritoneal lavage between healthy and septic WT and DEL-1 null mice. No effect of DEL-1 deficiency was noted on the above monocyte populations in neonates. These results are depicted in supplemental Fig. 2 of the revised manuscript.

Comment: 5. *Ex vivo phagocytosis assays of neutrophils impacted by Dell would further strengthen the conclusion of this report.*

Response: Thank you for this comment. To address whether DEL-1 affects phagocytosis in neonatal age, we performed ex vivo phagocytosis assay in neutrophils from whole blood. We infected neutrophils ex vivo, in the presence or absence of DEL-1-Fc, with unopsonised *E. coli* to recapitulate the phagocytosis conditions that occur in early life in the context of bacterial sepsis (as it is known that neonates are lacking opsonins, such as complement and IgG, and thus non-opsonic phagocytosis predominates in early life). We did not observe any difference in phagocytosis among adult WT or neonate WT mice treated with DEL-1-Fc compared with IgG-Fc. These results are now presented in supplemental Fig. 4 in the revised manuscript.

REVIEWER COMMENTS

Reviewer #1 (Remarks to the Author):

In this revised submission, the authors have addressed many of the concerns raised by the reviewers. Although the authors have chosen not to conduct several of the studies recommended by the reviewers, they have added additional data on phagocytosis, and have clarified several of the points raised. This reviewer believes that the authors are entitled to accept or reject the recommendations of the reviewers, and I am leery to make the same recommendations twice.

Survival to sepsis is a tenuous outcome to base a murine study on because of the differences in human and murine management. Much of the human response to sepsis is dependent on the management of the patient which cannot be controlled in the mouse. The utility of the mouse studies is always limited by not only the physiologic differences between the species, but how the two are managed during sepsis. This cannot be controlled.

1. Line 124. 34-37 week gestational infants are premature, but mortality is marginally if not increased. The real comparison is not at 34-37 weeks, but less than 32 weeks, or even better, less than 28 weeks where neonatal sepsis and NEC are elevated.

Reviewer #2 (Remarks to the Author):

Authors have addressed adequately all concerns raised.

Response to Reviewers

Manuscript Number: NCOMMS-21-42218

“A novel IL-10 – DEL-1 axis supports emergency granulopoiesis and sepsis survival in early life”

We would like to thank the reviewers for the constructive comments. Following the reviewer's suggestions, we have performed additional experiments to address all remaining concerns. All the reviewer's comments have been addressed point-by-point below.

Reviewer 1, comment 1: In this revised submission, the authors have addressed many of the concerns raised by the reviewers. Although the authors have chosen not to conduct several of the studies recommended by the reviewers, they have added additional data on phagocytosis, and have clarified several of the points raised. This reviewer believes that the authors are entitled to accept or reject the recommendations of the reviewers, and I am leery to make the same recommendations twice.

Response: We thank the reviewer for the response. In the first round, we took into serious consideration all the recommendations made. We addressed them all, either experimentally or by providing supporting evidence from previous studies. We are sorry that this response was not sufficient, and we therefore performed additional experiments to address all recommendations from round 1 as well.

From our understanding, the reviewer is referred to the following recommendation from round 1: *“The key question is how does DEL-1 regulate bone marrow myeloid cell accumulation and release. Does it act similar to SDF-1 (CXCL12); does it act through CSF and progenitors, or is the target MSCs. These mechanistic questions need consideration.”*

In the revised version, we further investigated how DEL-1 regulates bone marrow (BM) myeloid cell numbers. This involved an assessment of hematopoietic stem and progenitor cell numbers, the production of critical regulators of emergency myelopoiesis (e.g G-CSF, TNF- α , IL-6, IL-17, CXCL1) (PMID: 21224471, PMID: 34966386, PMID: 32849521, PMID: 34706233) and of factors that have been implicated in bone marrow retention, e.g. SDF-1, in WT and DEL-1 deficient mice, as suggested by the reviewer.

G-CSF is a major regulator of emergency myelopoiesis (PMID: 21224471). We examined whether G-CSF is altered in DEL-1 deficient neonatal mice compared to WT mice. We did not observe any alterations in the expression of G-CSF in serum in DEL-1 deficient neonatal mice compared to WT, upon steady state or septic conditions (supplementary Figure 9 in the revised manuscript).

While upregulation in the expression of inflammatory cytokines is an important event for promoting emergency myelopoiesis in the bone marrow, no difference was noted in the expression of inflammatory mediators (such as TNF, IL-6, IL-17, IL-10, CXCL1), in DEL-1-deficient neonatal mice exposed to polymicrobial sepsis as compared to WT controls (supplementary Figure 3 and 10 in the revised manuscript). Therefore, the defect in myelopoiesis that is noted in DEL-1 deficiency cannot be attributed to alterations in the expression of G-CSF or other inflammatory mediators that are known to regulate emergency myelopoiesis.

Since we observed a significantly reduced neutrophil pool in the bone marrow of DEL-1 deficient mice, we aimed to further characterize the role of DEL-1 in bone marrow granulocyte production and release. To assess whether the decreased numbers of granulocytes in the BM of DEL-1 deficient mice

were due to impaired retention of granulocytes in the BM, we analyzed SDF-1 expression. No difference was observed in the levels of SDF-1 upon sepsis (12 hours post cecal slurry sepsis) in the BM of neonatal mice due to DEL-1 deficiency (Supplementary figure 8). In addition to this, to further investigate neutrophil retention and release between BM and periphery in DEL-1 deficient neonatal mice, we have assessed neutrophil numbers in bone marrow and circulating neutrophils in peripheral blood simultaneously. Neutrophils decreased in bone marrow, while circulating neutrophils also decreased in the periphery (blood) (Figure 4c and figure 6c). We also observed that neutrophil numbers were significantly decreased over time in *Del1*^{-/-} neonatal mice compared to WT ones, both in the bone marrow and in the peripheral blood. Therefore, the effect of DEL-1 deficiency on the numbers of bone marrow neutrophils cannot be attributed to increased retention in the bone marrow.

Upon systemic microbial infection, emergency myelopoiesis entails activation of proliferation of hematopoietic progenitors in the bone marrow to compensate for the increased need for mature myeloid cells and restore bone marrow cellularity. To determine a functional role of DEL-1 in the regulation of emergency myelopoiesis, we performed here detailed bone marrow analysis in WT and DEL-1 deficient neonatal mice upon steady state and septic conditions. Although DEL-1 deficiency had no significant effect on long-term hematopoietic stem cells (LT-HSCs) and short-term HSCs (ST-HSCs) (Lin⁻cKit+Sca1+CD48⁻ CD150⁻) abundance, we observed a significant decrease in the number of multipotent progenitors (MPPs), and particularly the myeloid-biased MPP3 lineage, upon polymicrobial sepsis (Figure 7 in the revised manuscript) in DEL-1 deficient mice compared to WT mice. Besides MPP3 cells, we also observed a significant decrease of the percentage of granulocyte-macrophage progenitors (GMPs) (Lin⁻cKit+Sca1⁻CD16/32+CD34+) in the bone marrow of septic *Del1*^{-/-} neonates, compared to WT neonatal mice (Figure 8a, b in the revised manuscript). In the context of sepsis, DEL-1-Fc administration significantly prevented the decline of GMPs in the bone marrow of DEL-1 deficient neonate pups (Figure 8c in the revised manuscript).

Our above-discussed results are in agreement with previous studies (from the Chavakis and Hajishengallis labs; Mitroulis et al. J Clin Invest 2017; Chen et al. Thromb Haemost. 2018;118(03):613-616) showing that DEL-1, derived from endothelial and mesenchymal stromal cells in the bone marrow niche, promotes myelopoiesis predominantly under hematopoietic stress conditions in a juxtacrine manner, via integrin-dependent interactions with hematopoietic progenitors. In agreement with these findings, we show that *Del1*^{-/-} neonatal mice exhibited excessive depletion of the bone marrow multipotent progenitors. DEL-1 deficiency also resulted in a significant decrease in multipotent and myeloid-biased progenitors in the neonatal bone marrow under septic conditions, which affected mostly neutrophil numbers, without influencing other populations such as monocytes in the neonatal bone marrow. The above findings were independent of the levels of SDF-1, G-CSF or inflammatory cytokines. We also found that DEL-1 was elevated in the bone marrow of neonates compared to adults and was significantly induced in sepsis. These observations overall indicate that DEL-1 acts as a niche factor that supports myelopoiesis in the bone marrow upon systemic microbial infection in early life.

The above results as well as extensive discussion of the new findings have been added to the revised manuscript. In addition, since the effect of DEL-1 on progenitors in the bone marrow was only found upon sepsis and not in steady state and mostly granulopoiesis was affected, we modified the title of the article to point out the effect of DEL-1 in emergency granulopoiesis and not in myelopoiesis in general. We believe that the above results have expanded our understanding of the role of DEL-1 in emergency granulopoiesis particularly in early life.

Reviewer 1 - comment 2: *Survival to sepsis is a tenuous outcome to base a murine study on because of the differences in human and murine management. Much of the human response to sepsis is*

dependent on the management of the patient which cannot be controlled in the mouse. The utility of the mouse studies is always limited by not only the physiologic differences between the species, but how the two are managed during sepsis. This cannot be controlled.

Line 124. 34-37 week gestational infants are premature, but mortality is marginally if not increased. The real comparison is not at 34-37 weeks, but less than 32 weeks, or even better, less than 28 weeks where neonatal sepsis and NEC are elevated.

Response: These are valid points, and we thank the reviewer for pointing them out. In the revised manuscript, we modified our discussion in several points (highlighted in yellow) in order to clarify that the results refer to murine sepsis only. As far as extreme premature and very premature infants are concerned, it is not feasible for us to obtain a reasonable number of samples in due course from these populations, mostly due to low incidence of such extreme prematurity in our department. In addition, sampling collection from these newborns can be challenging due to their very low birth weight.

To discuss all concerns that the reviewer raised we added the following limitation paragraph in the discussion: *"However, our study has several limitations. The data on murine sepsis that we present in this study may not necessarily be relevant in the context of human sepsis. A great variety of confounding factors such as physiological and environmental factors as well as differences in patient management of septic humans cannot be controlled in murine models of sepsis. In addition, in this study, we mostly studied the expression and role of DEL-1 in term murine neonates, and we measured DEL-1 in the serum of late preterm (>34 weeks GA) or term human neonates. Term and late preterm neonates have higher mortality from sepsis compared to older children and adults, however we should acknowledge the highest susceptibility and mortality to sepsis in early life is noted in very preterm (28 - 32 weeks GA) or even extremely preterm (<28weeks GA) newborns, who were not examined in this study. Further studies are required to address the function and role of DEL-1 in human sepsis, especially in very preterm or extremely premature infants."*

We hope that we adequately stress out the limited relevance between murine and human sepsis as well as the importance of addressing the role of DEL-1 in even more preterm neonates.

Reviewer 2 (Remarks to the Author): Authors have addressed adequately all concerns raised.

Response: We cordially thank the reviewer for all previous comments that helped us improve our manuscript.

REVIEWERS' COMMENTS

Reviewer #1 (Remarks to the Author):

The authors have made a number of important additions to the study and have addressed other comments in their discussion. Although I have concerns about the validity of the model to recapitulate the human condition, particularly in neonates, I appreciate that this is my opinion, and consensus among the scientific community about mouse models in sepsis is controversial. But the authors have made a fair effort to either address my concerns experimentally or to adjust their discussion accordingly.

Response to Reviewers

Manuscript Number: NCOMMS-21-42218

We would like to thank the reviewers for the constructive comments. All the reviewer's comments have been addressed point-by-point below.

Reviewer Comment #1: The authors have made a number of important additions to the study and have addressed other comments in their discussion. Although I have concerns about the validity of the model to recapitulate the human condition, particularly in neonates, I appreciate that this is my opinion, and consensus among the scientific community about mouse models in sepsis is controversial. But the authors have made a fair effort to either address my concerns experimentally or to adjust their discussion accordingly.

Response: We cordially thank the reviewer for all critical previous comments that helped us improve our manuscript.

We agree with the reviewer that a murine model may not fully recapitulate human sepsis pathophysiology, especially in neonates. Murine models of sepsis have been a topic of considerable debate. Although the cecal slurry model is the preferred method of studying sepsis in neonatal mice (PMID: 34048005), it is well recognised that no single animal model can recapitulate the complex and varied clinical manifestations of the sepsis syndrome in humans (PMID: 32648582). There are clear differences between what is modelled in research animals and what is seen in patients. Some believe though that as long as correct interpretation of the data is being made, mice experiments can still be of scientific value, especially with regard to expanding the understanding of the underlying pathophysiology of sepsis, keeping in mind that mice and humans may not share common motifs and pathways and the potential validity of murine data to human needs to be further investigated (PMID: 32648582).

We respect the reviewer's opinion and for this reason, we avoid overstatements and clearly state the limitations of the work. We do thank the reviewer for raising all these important issues.